# Guiding antibiotics towards their target using bacteriophage proteins

Xinghong Zhao [1,2,6] ✉, Xinyi Zhong [1,2,6], Shinong Yang [1,2,6],
Jiarong Deng [1,2], Kai Deng [1,2], Zhengqun Huang [1,2], Yuanfeng Li [3],
Zhongqiong Yin [1] ✉, Yong Liu [4] ✉, Jakob H. Viel [5] & Hongping Wan [1,2] ✉

Novel therapeutic strategies against difficult-to-treat bacterial infections are desperately needed, and the faster and cheaper way to get them might be by repurposing existing antibiotics. Nanodelivery systems enhance the efficacy of antibiotics by guiding them to their targets, increasing the local concentration at the site of infection. While recently described nanodelivery systems are promising, they are generally not easy to adapt to different targets, and lack biocompatibility or specificity. Here, nanodelivery systems are created that source their targeting proteins from bacteriophages. Bacteriophage receptor-binding proteins and cell-wall binding domains are conjugated to nano-particles, for the targeted delivery of rifampicin, imipenem, and ampicillin against bacterial pathogens. They show excellent specificity against their targets, and accumulate at the site of infection to deliver their antibiotic payload. Moreover, the nanodelivery systems suppress pathogen infections more effectively than 16 to 32-fold higher doses of free antibiotics. This study demonstrates that bacteriophage sourced targeting proteins are promising candidates to guide nanodelivery systems. Their specificity, availability, and biocompatibility make them great options to guide the antibiotic nanodelivery systems that are desperately needed to combat difficult-to-treat infections.

The effectiveness of conventional antibiotics has been declining in the past decades due to the emergence of resistant bacteria[1–3]. Most alarming is the accelerated appearance of antibiotic resistance in the bacteria known as ESKAPE pathogens (*Enterococcus faecium*, *Staphylococcus aureus*, *Klebsiella pneumoniae*, *Acinetobacter baumannii*, *Pseudomonas aeruginosa*, and *Enterobacter* species)[4–6]. These bacterial strains carry antibiotic resistance genes and are highly virulent, causing life threatening infections[7]. As a result, important medical treatments that rely on antibiotics, like organ transplants, chemotherapy, or prevention of co-infection, are predicted to become riskier and less successful in the future[8,9].

Unfortunately, the number of newly approved first-in-class antibiotics has been steadily decreasing in the past two decades, especially those for the treatment of infections caused by Gram-negative pathogens[10–14]. This decrease can be partially explained by the financial risk of developing novel antibiotics[15]. To circumvent this problem, researchers are looking for strategies that enhance existing antibiotics, rather than developing completely new therapies. One such strategy is the use of targeted nanodelivery systems.

[1]Center for Sustainable Antimicrobials, Department of Pharmacy, College of Veterinary Medicine, Sichuan Agricultural University, Chengdu 611130, China. [2]Center for Infectious Diseases Control (CIDC), College of Veterinary Medicine, Sichuan Agricultural University, Chengdu 611130, China. [3]Translational Medicine Laboratory, The First Affiliated Hospital of Wenzhou Medical University, Wenzhou, Zhejiang 325035, China. [4]Wenzhou Institute, University of Chinese Academy of Sciences, Wenzhou, Zhejiang 325001, China. [5]Groningen Biomolecular Sciences and Biotechnology Institute, University of Groningen, 9747AG Groningen, Netherlands. [6]These authors contributed equally: Xinghong Zhao, Xinyi Zhong, Shinong Yang. ✉e-mail: xinghong.zhao@sicau.edu.cn; yinzhongq@163.com; y.liu@ucas.ac.cn; hpwan@sicau.edu.cn

Nanodelivery systems combine antibiotics with high affinity target-binding. When targeting bacteria, they have been reported to significantly enhance the therapeutic efficacy of antibiotics[16–19]. Rifalogue, a rifampicin derivative conjugated to an *S. aureus*-specific antibody, performed better than the free antibiotic in treating systemic methicillin-resistant *S. aureus* (MRSA) infections in a mouse model[20]. While this is highly impressive, the large-scale application of antibody conjugates is hindered by production cost, as each Rifalogue molecule requires an expensive antibody to produce. Moreover, while antibodies are highly specific, they are also sensitive to e.g., pH, temperature, and bacterial proteases[21,22]. These limitations could complicate their use in the complex microenvironment of bacterial infection sites.

More recently, a non-antibody nanodelivery system with high loading capacity for antibiotics was developed against *S. aureus*[23]. The cyclic 9-amino acid peptide CARG was engineered to have high affinity to *S. aureus* by employing a phage display-based high-throughput screening system in an *S. aureus*-induced pneumonia model[23]. CARG was then used to coat the surface of biocompatible porous silicon nanoparticles (pSiNPs). When loaded with vancomycin, CARG-conjugated pSiNPs (CARG-pSiNPs) have significantly enhanced therapeutic efficacy compared to the free antibiotic in an *S. aureus*-induced pneumonia model[23].

While the CARG-pSiNPs-conjugate successfully addresses the problem of payload per conjugate molecule, it is a considerable challenge to engineer such cyclic peptides with affinity to other bacterial pathogens in similar fashion. Development of these molecules might be further impeded by the small size of the cyclic peptide, limiting the freedom of design in specificity and affinity engineering. Recently, the use of proteins from bacteriophages (phages) is being explored as an alternate strategy to obtain molecules with high affinity to bacterial targets[24–26].

Phages are bacteria-specific viruses that infect their hosts with high specificity[27,28]. For the successful infection of a host, they produce receptor-binding proteins (RBPs), which are responsible for the phage-host recognition and interaction[24]. RBPs show a comparable or even superior specificity and affinity to bacterial host than antibodies[24,29,30].

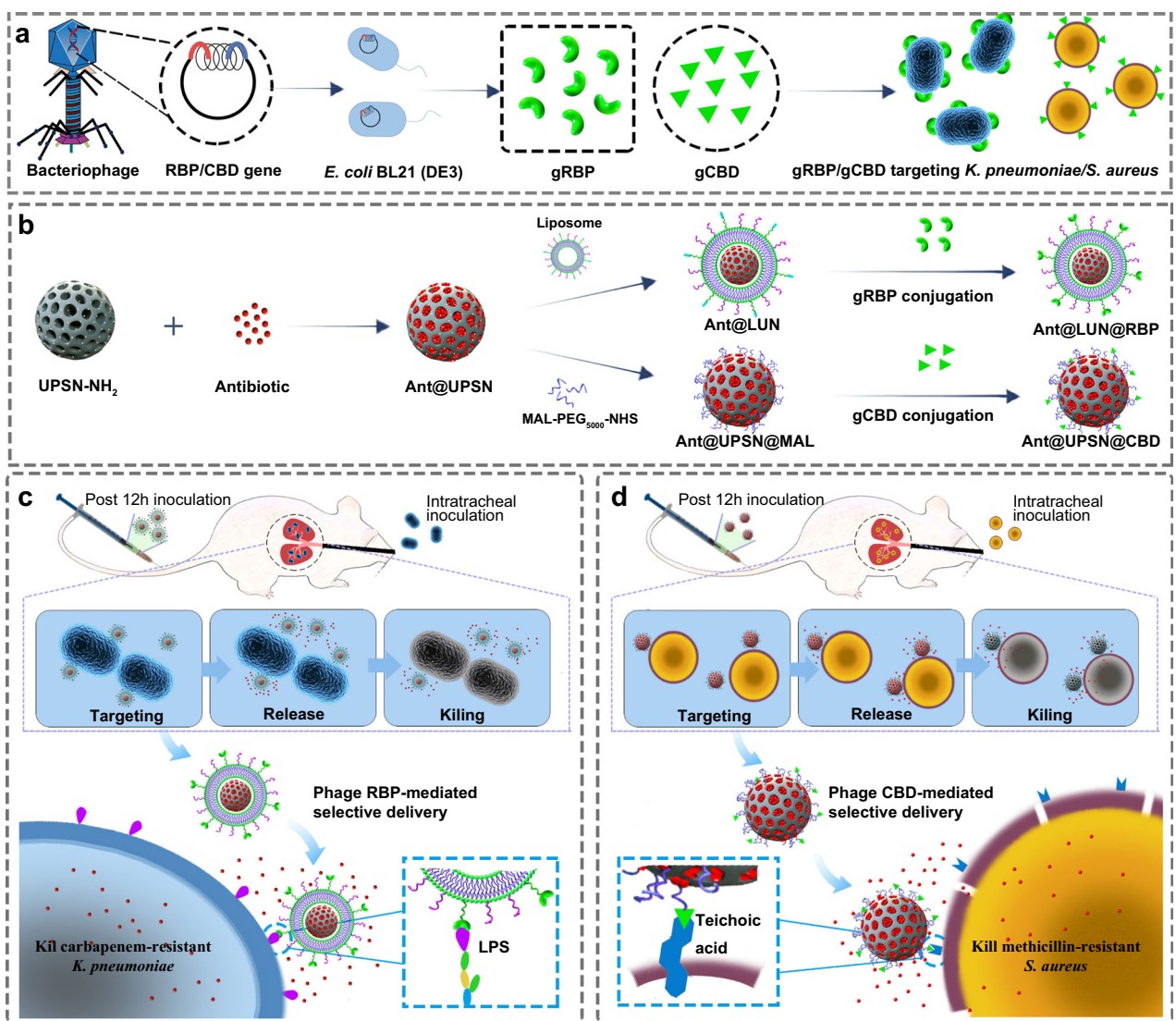

**Fig. 1 | Schematic illustrations of the construction of bacterial targeted antibiotic nanodelivery systems. a** Heterologous expression of targeting proteins by employing bacteriophage origin RBPs and CBDs. **b** Development of two distinct nanodelivery systems by employing heterologous expressed RBP and CBD as targeting devices: lipid-coated UPSNs (LUN) bearing RBPs (LUN@RBP) and CBDs modified UPSNs (UPSN@CBD). **c** In a CRKP-induced mouse pneumonia model, antibiotic-loaded LUN@RBP (Ant@LUN@RBP) more effectively suppressed CRKP infections than untargeted antibiotic nanoparticles or of free antibiotics. **d** In an MRSA-induced mouse pneumonia model, antibiotic-loaded UPSN@CBD (Ant@UPSN@CBD) more effectively suppressed MRSA infections than untargeted antibiotic nanoparticles and free antibiotics.

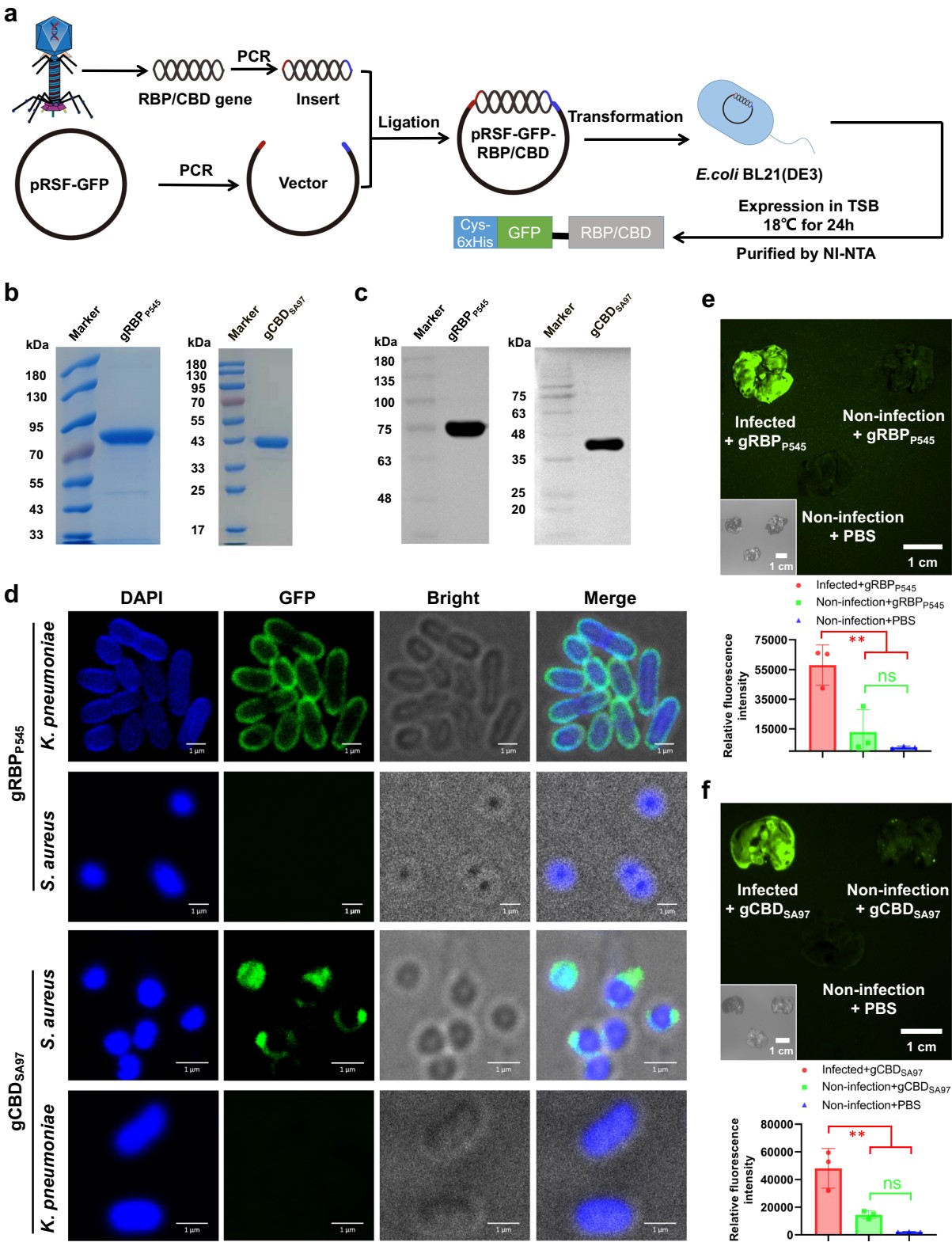

Additionally, phages of Gram-positive bacteria can produce endolysins, bacterial cell wall peptidoglycan hydrolases which destabilize the host's cell wall for the release of progeny virus[31–34]. Phage endolysins are guided by a cell-wall binding domain (CBD) that leads the catalytic domain to the bacterial host peptidoglycan via high-affinity non-covalent binding[24,32,35,36].

Both RBPs and CBDs have great specificity and affinity to their bacterial hosts. Moreover, there is a natural abundance of host-specific

phages[25,37–39], and the proteinaceous nature of RBPs and CBDs make them relatively easy to engineer. Taking the previous into account, phage-protein antibiotic conjugates are highly interesting candidates for bacteria-targeted antibiotic nanodelivery systems.

In this study, RBPs and CBDs were applied to develop two distinct high-efficacy nanodelivery systems for the targeted delivery of antibiotics to two respective ESKAPE pathogens. First, a *K. pneumoniae* targeted antibiotic nanodelivery system was developed by employing

**Fig. 2 | Heterologous expressed gRBP$_{P545}$ and gCBD$_{SA97}$ show selective binding to cultured pathogenic bacteria in vitro and home to pathogenic bacteria-infected lungs in vivo. a** Schematic representation of the heterologous expression of gRBP$_{P545}$ and gCBD$_{SA97}$. SDS-PAGE Images (**b**) and anti-His6 western blot (**c**) of the heterologously expressed gRBP$_{P545}$ and gCBD$_{SA97}$. Three times the experiment was repeated with similar results. **d** Confocal laser scanning microscopy images of CRKP and MRSA after incubation with gRBP$_{P545}$ and gCBD$_{SA97}$ (green). Pathogenic bacteria are visualized under a phase contrast model, and bacterial nucleoid is stained with DAPI (blue). Three times the experiment was repeated with similar results. **e** Time-gated fluorescence images of gRBP$_{P545}$ in lungs harvested from mice after 30 min of circulation. *K. pneumoniae*-induced lung infection was generated by intratracheal inoculation of CRKP. At 24 h post-infection, gRBP$_{P545}$ was intravenously injected and allowed to circulate for 30 min. After that, lungs were harvested for time-gated fluorescence imaging using a FUSION FX7 EDGE Imaging System.

Mice without CRKP infection were treated with the same dose of gRBP$_{P545}$ or the same volume of PBS as controls. Data are presented as mean ± standard deviation ($n = 3$ biological replicates). The statistical significance of the data was assessed using one-way ANOVA followed by Tukey's multiple comparisons test. ns, no significance; **$p < 0.001$. **f** Time-gated fluorescence image of gCBD$_{SA97}$ in lungs harvested from mice after 30 min of circulation. *S. aureus*-induced lung infection was generated by intratracheal inoculation of MRSA. At 24 h post-infection, gCBD$_{SA97}$ was intravenously injected and allowed to circulate for 30 min. After that, lungs were harvested for time-gated fluorescence imaging using a FUSION FX7 EDGE Imaging System. Mice without MRSA infection were treated with the same dose of gCBD$_{SA97}$ or the same volume of PBS as controls. Data are presented as mean ± standard deviation ($n = 3$ biological replicates). The statistical significance of the data was assessed using one-way ANOVA followed by Tukey's multiple comparisons test. ns, no significance; **$p < 0.001$. Source data are provided as a Source Data file.

*K. pneumoniae* phage RBPs. For this delivery system, biocompatible urchin-like porous silica nanoparticles (UPSNs) were used as the core, providing high loading capacity for antibiotics[40,41]. After loading the UPSNs with antibiotics, they were coated with a lipid bilayer to allow for intracellular delivery of the antibiotic. In a final step, *K. pneumoniae* phage RBPs were conjugated to the lipid bilayer (Fig. 1). Second, a similar nanodelivery system was developed against *S. aureus*, now employing *S. aureus* phage endolysin CBDs, to guide the antibiotic to the target cell surface (Fig. 1). Both systems significantly enhanced the therapeutic efficacy of antibiotics in respectively carbapenem-resistant *K. pneumoniae* (CRKP) and MRSA-induced mouse pneumonia models. And, by doing so, demonstrate the effectiveness of phage derived antibiotic nanodelivery systems against ESKAPE pathogens.

## Results and discussion
### Creation and verification of the targeting module
To develop phage protein-guided antibiotic nanodelivery systems, we conjugated heterologously expressed phage-host interaction proteins RBP and CBD to a UPSN delivery module respectively[24,25,42]. In order to provide a proof of principle for both Gram-positive and Gram-negative pathogens, RBP from *K. pneumoniae* phage P545 (RBP$_{P545}$) and endolysin CBD from *S. aureus* phage SA97 (CBD$_{SA97}$) were selected as targeting proteins[42,43]. First, the genes encoding RBP$_{P545}$ and CBD$_{SA97}$ (Supplementary Table 1) were genetically fused to a 6xHis-tag-GFP protein, allowing for the heterologous expression of 6xHis-GFP-RBP$_{P545}$ (gRBP$_{P545}$, Supplementary Table 2) and 6xHis-GFP-CBD$_{SA97}$ (gCBD$_{SA97}$, Supplementary Table 2) in *Escherichia coli* BL21(DE3) (Fig. 2a). Additionally, an N-terminal cysteine residue was genetically introduced to both constructs to facilitate their conjugation to the delivery module via maleimide-mediated bioconjugation[23,44]. After expression and purification, the obtained protein sizes were determined by sodium dodecyl sulfate-polyacrylamide gel electrophoresis (SDS-PAGE) and western blot to be the expected 77 kDa and 41 kDa respectively (Fig. 2b, c). Both analyses also showed high purity for both proteins.

After purification, the binding capacity of gRBP$_{P545}$ and gCBD$_{SA97}$ was verified by respectively incubating them with different *K. pneumoniae* and *S. aureus* strains (Supplementary Table 3). Analysis by fluorescence microscopy and confocal laser scanning microscopy (CLSM) showed that gRBP$_{P545}$ and gCBD$_{SA97}$ bonded to all intended target strains (Fig. 2d, Supplementary Figs. 1, 2). Additionally, both proteins were tested against five non-target strains, for which they showed no measurable affinity (Supplementary Figs. 3, 4). These results demonstrate an exceptionally broad range specificity within the target strain, while maintaining a narrow range spectrum. This is in line with the previous findings that RBP and CBD have a broader affinity spectrum than the fully assembled phages from which they are sourced[26,42,45,46].

After confirming that both gRBP$_{P545}$ and gCBD$_{SA97}$ have excellent binding capacity and specificity towards their intended targets in vitro, their targeting capacity was also tested in vivo. To do this, gRBP$_{P545}$

and gCBD$_{SA97}$ were applied in CRKP and MRSA-induced mouse pneumonia models, respectively. Time-gated fluorescence imaging of lung tissue harvested from these mouse models showed that gRBP$_{P545}$ was significantly accumulated in the lungs of CRKP-infected mice, and gCBD$_{SA97}$ was significantly accumulated in the lungs of mice infected with MRSA, while accumulation of either protein was absent in non-infected mice (Fig. 2e, f). This result is a very promising confirmation that the targeting proteins can still bind their bacterial targets in vivo. After this result, the antibiotic loading modules could be prepared for conjugation.

### Creation and characterization of the antibiotic delivery systems
For creation of the antibiotic loading module (Fig. 3a and Supplementary Fig. 5), UPSNs were selected as the core because of their excellent biocompatibility and high loading capacity[47-51]. The UPSN cores were synthesized using a cetyltrimethylammonium bromide template, and a tetraethyl orthosilicate precursor[52]. Analysis of the synthesized particles by transmission electron microscope (TEM) showed that uniform UPSNs were successfully obtained (Fig. 3b). The synthesized UPSNs have a hydrodynamic diameter of 161.0 ± 5.2 nm and a zeta potential of −23.0 ± 1.5 mV, measured by dynamic light scattering (DLS) (Fig. 3c, d, Supplementary Fig. 6, Supplementary Table 4).

After confirming that the UPSNs were successfully synthesized, they were aminated using (3-aminopropyl)triethoxysilane resulting in UPSN-NH$_2$ (Fig. 3b–d, Supplementary Fig. 6, Supplementary Table 4). After amination, the UPSNs were loaded with rifampicin, which was confirmed to be successful by a measured decrease in zeta potential and through TEM imaging (Fig. 3b–d, Supplementary Fig. 6, Supplementary Table 4). A high loading capacity of about 60% rifampicin by mass was achieved (Supplementary Table 5), which could be attributed to the high porosity of UPSN compared to that of previously reported mesoporous silica nanoparticles[23,53].

The rifampicin-loaded UPSN (Rif@UPSN) was then used as the core of the nanodelivery system Rif@UPSN@CBD$_{SA97}$. In an alternative synthesis pathway, Rif@UPSN was enveloped by a liposome to create Rif@LUN, which was used as the core of nanodelivery system Rif@LUN@RBP$_{P545}$.

To create Rif@UPSN@CBD$_{SA97}$, a bifunctional polyethylene glycol (PEG) [Poly(ethylene glycol) (N-hydroxysuccinimide 5-pentanoate) ether N′-(3-maleimidopropionyl)aminoethane, NHS-PEG$_{5000}$-MAL] was attached to the UPSN-NH$_2$ through an amine-NHS reaction, which was further linked to gCBD$_{SA97}$ through a MAL-thiol reaction (Fig. 3a, Supplementary Fig. 5). Compared to Rif@UPSN, Rif@UPSN@CBD$_{SA97}$ showed a slightly increased hydrodynamic diameter (183.3 ± 1.5 nm) and a decreased zeta potential (−5.2 ± 0.7 mV) (Fig. 3c, d and Supplementary Table 4), suggesting the successful modification of CBD$_{SA97}$.

In Rif@LUN@RBP$_{P545}$, the liposome consisting of 1,2-distearoyl-sn-glycero-3-phosphoethanolamine-N-[amino(polyethylene glycol)−2000] (DSPE-PEG$_{2000}$) and 1,2-distearoyl-sn-glycero-3-phosphoethanolamine-N-[maleimide(polyethylene glycol)−2000] (DSPE-PEG$_{2000}$-MAL) was

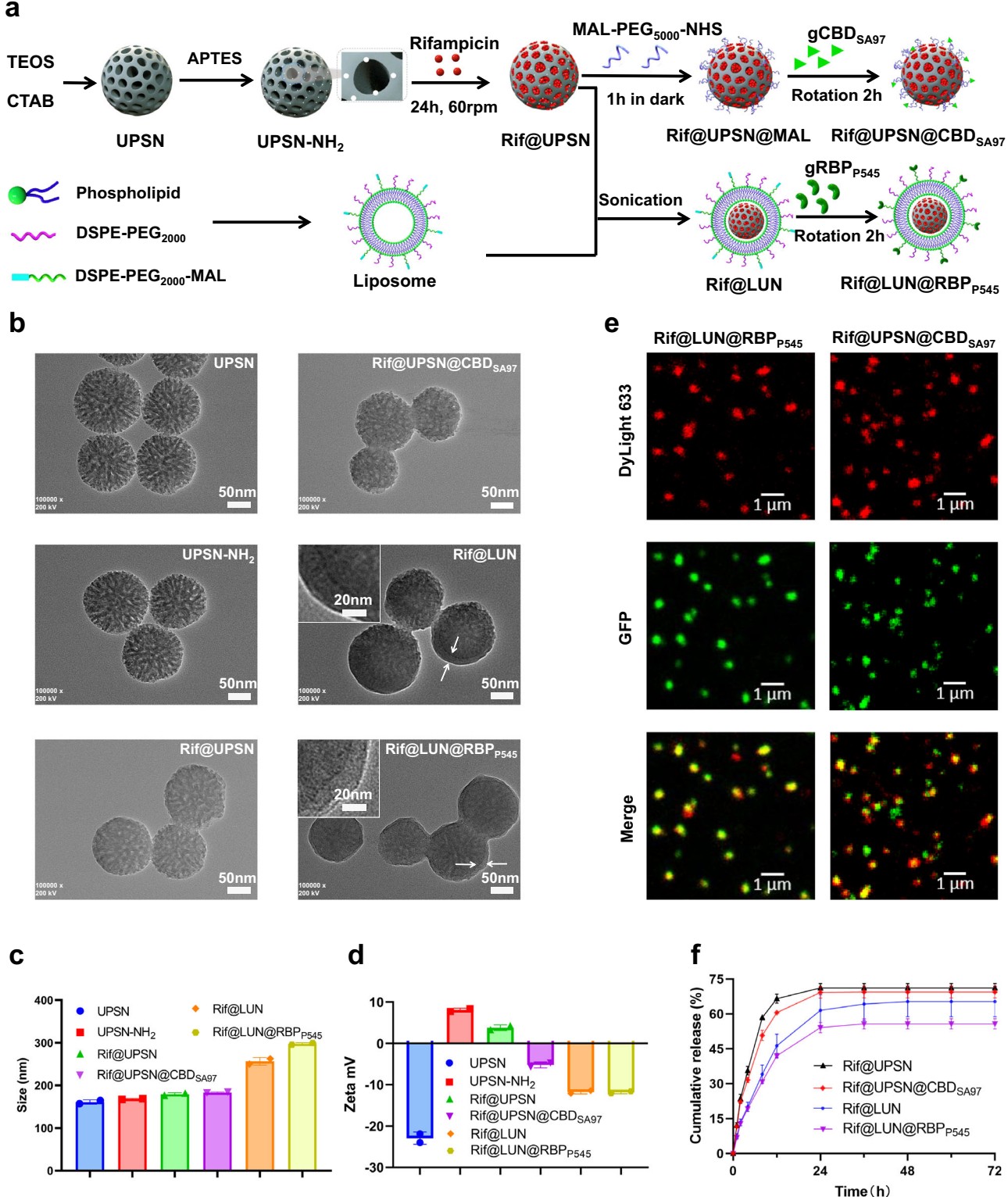

**Fig. 3 | Characterization of antibiotic-loaded nanodelivery systems.**
**a** Preparation routes of Rif@LUN@RBP$_{P545}$ and Rif@UPSN@CBD$_{SA97}$.
**b** Transmission electron microscope images of UPSN, UPSN-NH$_2$, Rif@UPSN, Rif@UPSN@CBD$_{SA97}$, Rif@LUN, and Rif@LUN@RBP$_{P545}$. **c** Average hydrodynamic size of UPSN, UPSN-NH$_2$, Rif@UPSN, Rif@UPSN@CBD$_{SA97}$, Rif@LUN, and Rif@LUN@RBP$_{P545}$ measured by dynamic light scattering. Data are presented as mean ± standard deviation ($n = 2$ independent experiments). **d** Surface zeta-potential of UPSN, UPSN-NH$_2$, Rif@UPSN, Rif@UPSN@CBD$_{SA97}$, Rif@LUN, and

Rif@LUN@RBP$_{P545}$ in ultrapure water. Data are presented as mean ± standard deviation ($n = 2$ independent experiments). **e** Confocal laser scanning microscopy images of Rif@LUN@RBP$_{P545}$ and Rif@UPSN@CBD$_{SA97}$ in which USPN was labeled with DyLight 633 (red) and the targeting devices were fused with GFP (green).
**f** Release profiles of rifampicin payload from the nanoparticles in PBS at 37 °C. Data are presented as mean ± standard deviation ($n = 3$ independent experiments). Source data are provided as a Source Data file.

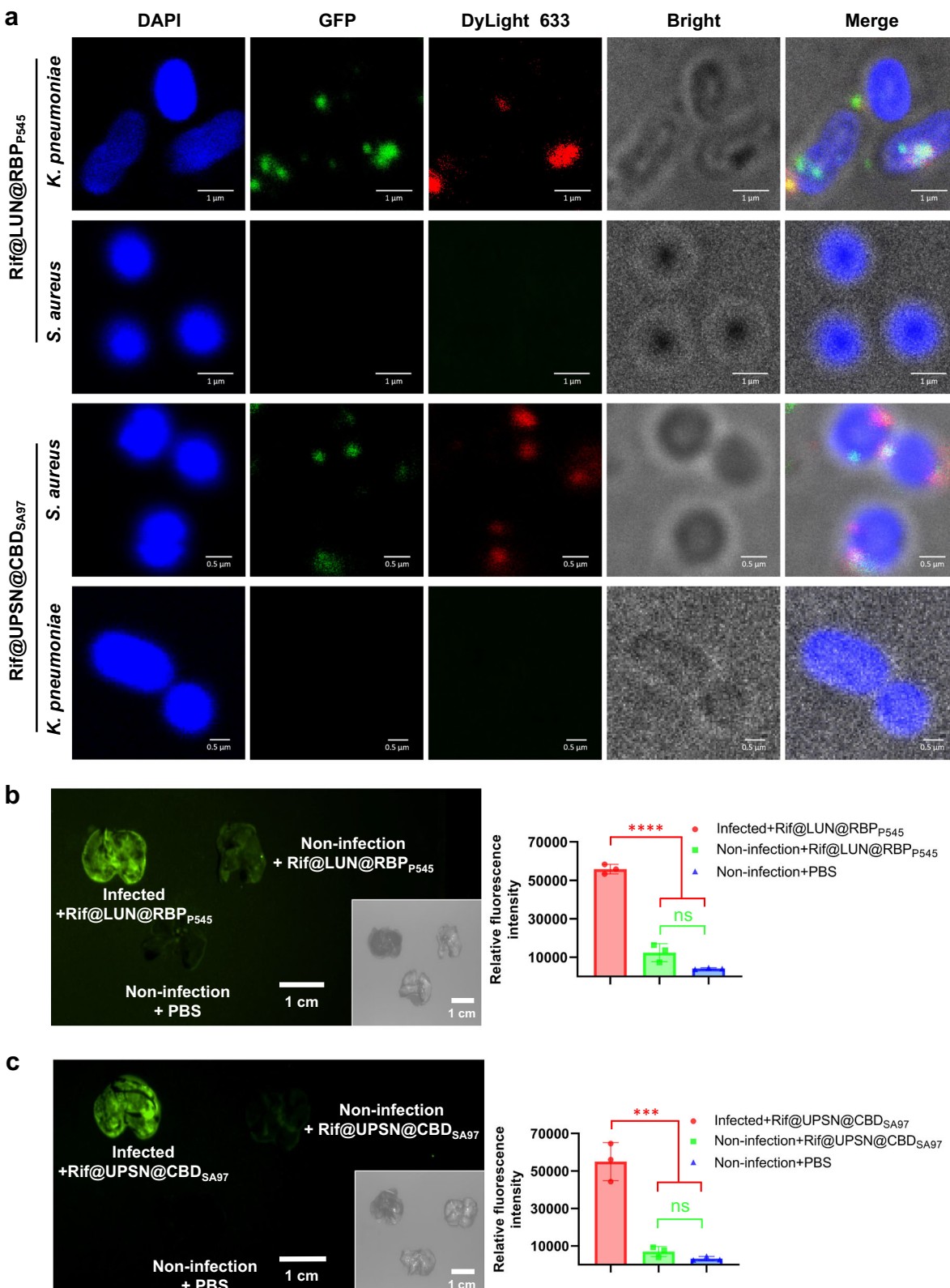

firstly coated to the surface of Rif@UPSN via electrostatic force by sonication, forming Rif@LUN, and then the thiol residue containing gRBP$_{P545}$ was conjugated to Rif@LUN via MAL-thiol reaction (Fig. 3a, Supplementary Fig. 5). The increased hydrodynamic diameters and decreased zeta potentials of Rif@LUN@RBP$_{P545}$ and Rif@LUN with respect to Rif@UPSN imply the successful liposomal coating (Fig. 3c,d

and Supplementary Table 4), which was further identified by TEM visualization (Fig. 3b).

The presence of gCBD$_{SA97}$ and gRBP$_{P545}$ in the antibiotic-loaded nanodelivery systems, in which UPSNs were labeled with DyLight 633 (red), was confirmed by CLSM (Fig. 3e). Quantity of gCBD$_{SA97}$ and gRBP$_{P545}$ on the nano vehicles was verified by a bicinchoninic acid

**Fig. 4 | Antibiotic-loaded nanodelivery systems selectively bind to pathogenic bacteria and precisely target the sites of infection. a** Confocal laser scanning microscopy images of CRKP and MRSA after incubation with Rif@LUN@RBP$_{545}$ and Rif@UPSN@CBD$_{SA97}$ in which USPN was labeled with DyLight 633 (red) and the targeting devices were fused with GFP (green). Pathogenic bacteria are visualized under a phase contrast model, and bacterial nucleoid is stained with DAPI (blue). Three times the experiment was repeated with similar results. **b** Time-gated fluorescence image of Rif@LUN@RBP$_{545}$, in which RBP$_{545}$ was fused with GFP (green), in lungs harvested from mice after 30 min of circulation. *K. pneumoniae*-induced lung infection was generated by intratracheal inoculation of CRKP. At 24 h post-infection, Rif@LUN@RBP$_{545}$ was intravenously injected and allowed to circulate for 30 min. After that, lungs were harvested for time-gated fluorescence imaging using a FUSION FX7 EDGE Imaging System. Mice without CRKP infection were treated with the same dose of Rif@LUN@RBP$_{545}$ or the same volume of PBS as

controls. Data are presented as mean ± standard deviation (*n* = 3 biological replicates). The statistical significance of the data was assessed using one-way ANOVA followed by Tukey's multiple comparisons test. ns, no significance; ****$p$ < 0.001. **c** Time-gated fluorescence image of Rif@UPSN@CBD$_{SA97}$, in which CBD$_{SA97}$ was fused with GFP (green), in lungs harvested from mice after 30 min of circulation. *S. aureus*-induced lung infection was generated by intratracheal inoculation of MRSA. At 24 h post-infection, Rif@UPSN@CBD$_{SA97}$ was intravenously injected and allowed to circulate for 30 min. After that, lungs were harvested for time-gated fluorescence imaging using a FUSION FX7 EDGE Imaging System. Mice without MRSA infection were treated with the same dose of Rif@UPSN@CBD$_{SA97}$ or the same volume of PBS as controls. Data are presented as mean ± standard deviation (*n* = 3 biological replicates). The statistical significance of the data was assessed using one-way ANOVA followed by Tukey's multiple comparisons test. ns, no significance; ***$p$ < 0.001. Source data are provided as a Source Data file.

assay. The results showed that moderate contents of gCBD$_{SA97}$ (2.7 ± 0.2 nmol/mg) and gRBP$_{545}$ (2.1 ± 0.1 nmol/mg) were presented in Rif@UPSN@CBD$_{SA97}$ and Rif@LUN@RBP$_{545}$, respectively (Supplementary Table 6).

After confirming the correct structure of the engineered nanodelivery systems, the ability of these systems to release their rifampicin payload was investigated (Fig. 3f). The non-coated Rif@UPSN and Rif@UPSN@CBD$_{SA97}$ showed rifampicin release curves of about 70% in 24 h. The delivery systems with a liposomal coating, Rif@LUN and Rif@LUN@RBP$_{545}$, released a lower but still functional ≈60% and ≈55% of their rifampicin payload in that time, respectively.

## The antibiotic-loaded nanodelivery systems target the sites of infection

To assess the pathogen targeting capability of the rifampicin-loaded nanodelivery systems, CLSM assays with DyLight-633-labeled Rif@LUN@RBP$_{545}$ and Rif@UPSN@CBD$_{SA97}$, were performed on DAPI-stained CRKP and MRSA. The microscopy images show that efficient binding of Rif@LUN@RBP$_{545}$ occur in CRKP, but not in *S. aureus* (Fig. 4a). Similarly, Rif@UPSN@CBD$_{SA97}$ showed specific binding to MRSA, but not against *K. pneumoniae* (Fig. 4a). These results show that Rif@LUN@RBP$_{545}$ and Rif@UPSN@CBD$_{SA97}$ have the intended bacterial targeting ability and specificity, suggesting they may exert efficient site-of-infection pathogen targeting capacity.

After verifying the bacterial targeting ability of Rif@LUN@RBP$_{545}$ and Rif@UPSN@CBD$_{SA97}$ in vitro, time-gated fluorescence imaging was performed on lung tissue of infected mice, after administering the delivery systems. The results showed that Rif@LUN@RBP$_{545}$ was significantly accumulated in the lungs of CRKP-infected mice, while no trend of Rif@LUN@RBP$_{545}$ accumulation was observed in the lungs of non-infected mice (Fig. 4b). Similarly, Rif@UPSN@CBD$_{SA97}$ was significantly accumulated in the lungs of MRSA-infected mice, while no trend of Rif@UPSN@CBD$_{SA97}$ accumulation was observed in the lungs of non-infected mice (Fig. 4c). These results suggest that RBP$_{545}$-guided Rif@LUN@RBP$_{545}$ and CBD$_{SA97}$-guided Rif@UPSN@CBD$_{SA97}$ may show enhanced therapeutic efficacy relative to equivalent doses of free rifampicin in vivo.

## LUN@RBPP545 and UPSN@CBDSA97 are biocompatible

All previous characterization steps of the presented nanodelivery systems show promising results. However, for their application as targeted antibiotics in the treatment of *K. pneumoniae* and *S. aureus* infections, LUN@RBP$_{545}$ and UPSN@CBD$_{SA97}$ should be biocompatible. For this purpose, cytotoxicity, hemolytic activity, and in vivo toxicity assays were performed on LUN@RBP$_{545}$ and UPSN@CBD$_{SA97}$[53–58]. For cytotoxicity assays, the influence of LUN@RBP$_{545}$ and UPSN@CBD$_{SA97}$ on the cell viability was assessed on two cell lines, hepatoblastoma cell line (Hep G2) and human embryonic kidney 293 T (HEK-293T). The results showed that both LUN@RBP$_{545}$ and UPSN@CBD$_{SA97}$ had no significant (*P* > 0.05) influence on the cell viabilities of Hep G2 and HEK-293T at a

concentration range of 16 to 512 µg/mL (Fig. 5a, b), demonstrating no concerns of cytotoxicity for LUN@RBP$_{545}$ and UPSN@CBD$_{SA97}$. Correspondingly, hemolytic activity was not observed for either molecule at a concentration range of 16 to 512 µg/mL (Fig. 5c).

To substantiate these results, a comprehensive analysis of blood chemistry and major blood cell populations was conducted 7 days post administration in the mouse model (Fig. 5d, e). Compared to the mice administered with only phosphate-buffered saline (PBS) medium, no significant difference was observed for all investigated parameters. In addition, there were no significant differences in body weight changes between the treated groups and untreated control group (Fig. 5f). Subsequent analysis by haematoxylin and eosin staining of the heart, liver, spleen, lungs, and kidneys of the treated and untreated group revealed no major changes (Fig. 5g). All these results demonstrate that there are no signs of acute toxicity of LUN@RBP$_{545}$ and UPSN@CBD$_{SA97}$, and confirm previous reports about the biosafety of porous silica nanoparticles, phages, and phage endolysins respectively[48,53,59–62]. Together, these experiments demonstrated a favorable safety profile for the nanodelivery systems, LUN@RBP$_{545}$ and UPSN@CBD$_{SA97}$, and their therapeutic efficacy could be assessed.

## LUN@RBPP545 and UPSN@CBDSA97 are re-applicable

Previous studies have shown that in vivo use of bacteriophages can cause an immune response[63–65], which may result in a decreased therapeutic efficacy upon repeated administration. For this reason, it was tested whether LUN@RBP$_{545}$ and UPSN@CBD$_{SA97}$ elicited immune responses that could cause any therapeutically undesirable events. After LUN@RBP$_{545}$ was administrated to mice, their immune response was probed by an enzyme linked immunosorbent assay (ELISA). This assay showed that RBP$_{545}$-specific antibodies, IgG and IgM, were produced (Fig. 6a, b). Similarly, CBD$_{SA97}$-specific antibodies, IgG and IgM, were produced following treatment with UPSN@CBD$_{SA97}$ (Fig. 6a, b). No RBP$_{545}$-specific or CBD$_{SA97}$-specific IgA was detected (Fig. 6c), which was expected since IgA is produced in mucosal immunization.

To assess whether the produced specific antibodies could influence the infection site targeting efficacy of LUN@RBP$_{545}$ and UPSN@CBD$_{SA97}$, time-gated fluorescence imaging assays were performed on lung tissues of infected mice following administration of LUN@RBP$_{545}$ and UPSN@CBD$_{SA97}$. The results show that LUN@RBP$_{545}$ still significantly accumulated in the lungs of LUN@RBP$_{545}$ immunized mice after CRKP inoculation (Fig. 6d). Notably, no significant differences in lung fluorescent intensity were observed between LUN@RBP$_{545}$ immunized and PBS-treated mice upon CRKP infection (Fig. 6d). Similar results were obtained for UPSN@CBD$_{SA97}$, which significantly accumulated in the lungs of UPSN@CBD$_{SA97}$ immunized mice upon MRSA infection (Fig. 6e). Also here, no significant differences in lung fluorescent intensity were observed (Fig. 6e). These results suggest that the infection site targeting capabilities of LUN@RBP$_{545}$ and UPSN@CBD$_{SA97}$ are not lower

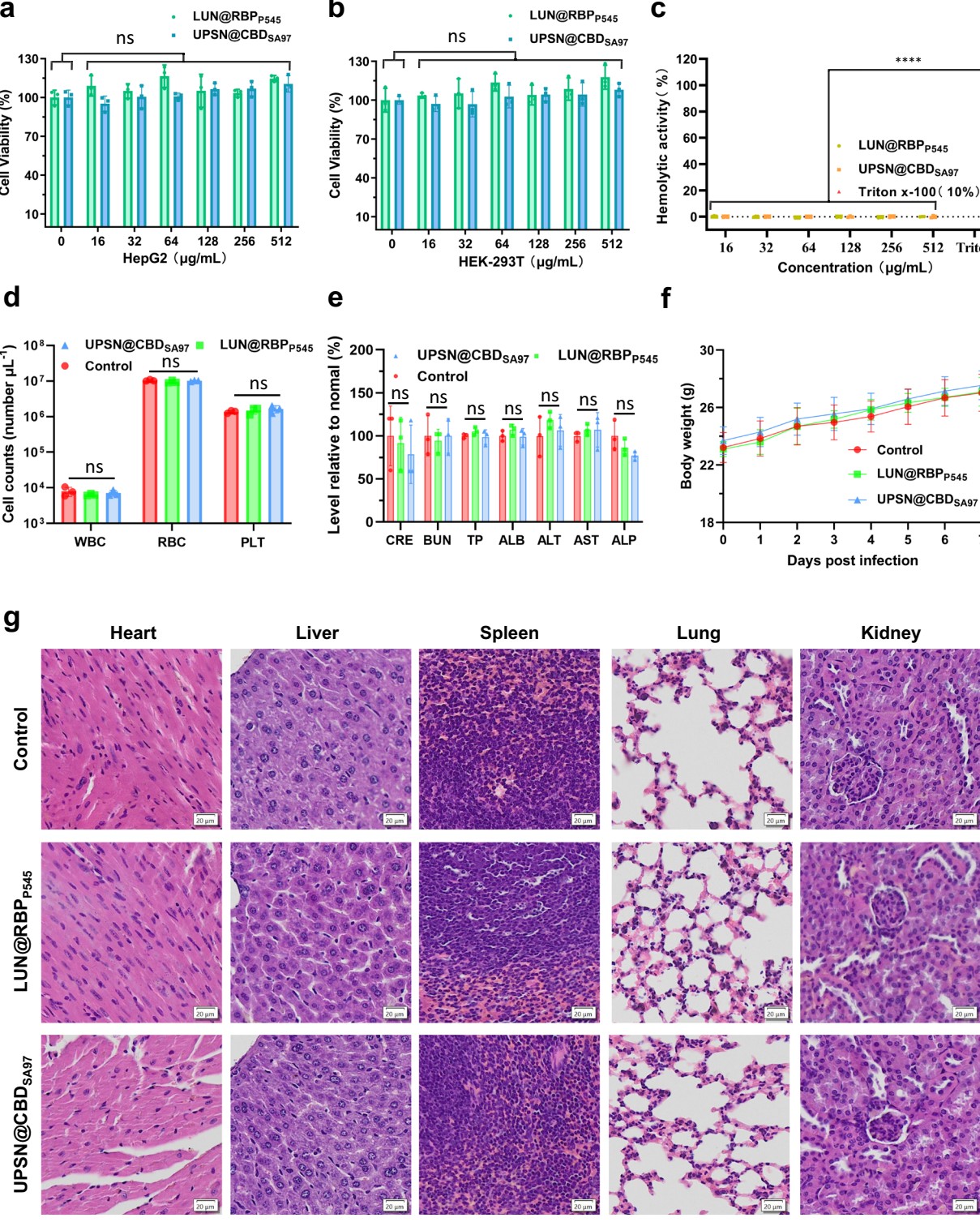

upon a third administration. And, while the effects of even more repetitions in the application are yet to be determined, these results indicate that the engineered nanodelivery systems, LUN@RBP_P545 and UPSN@CBD_SA97, can be applied at least thrice without losing efficacy.

## The therapeutic efficacy of rifampicin is enhanced by the nanodelivery systems

After confirmation of the biosafety and reapplication property of LUN@RBP_P545 and UPSN@CBD_SA97, the ability of Rif@LUN@RBP_P545 and Rif@UPSN@CBD_SA97 to treat acute lung infections in vivo was examined. The infection model for assessing the therapeutic efficacy

of Rif@LUN@RBP_P545 was created by intratracheally introducing CRKP, causing serious pneumonia. Untreated, this resulted in 80% mortality ($n = 10$ mice) 12–48 h post-infection (Fig. 7a, b and Supplementary Fig. 7). Treatment of the infected mice with intravenous injections of Rif@LUN@RBP_P545 at a dose of 2.5 mg/kg 12 h post-infection resulted in 90% recovery and long-term survival (Fig. 7b), whereas only 20% survival was observed when an identical dose by weight of free rifampicin was administered (Supplementary Fig. 7). Moreover, administration of rifampicin at a dose of 40 mg/kg was required to equal the 90% recovery achieved by 2.5 mg/kg Rif@LUN@RBP_P545 (Supplementary Fig. 7), demonstrating a 16-fold

**Fig. 5 | Biosafety evaluation of LUN@RBP$_{P545}$ and UPSN@CBD$_{SA97}$.** Viability of Hep G2 (**a**) and HEK-293T (**b**) after treatment with LUN@RBP$_{P545}$ or UPSN@CBD$_{SA97}$ at concentrations ranging from 16 to 512 µg/mL for 24 h. Data are presented as mean ± standard deviation ($n = 3$ biological replicates). The statistical significance of the data was assessed using one-way ANOVA followed by Tukey's multiple comparisons test. ns, no significance, vs. untreated cells. **c** Rabbit erythrocytes were incubated with LUN@RBP$_{P545}$ or UPSN@CBD$_{SA97}$ at concentrations ranging from 16 to 512 µg/mL. Their hemolytic activity was assessed by the release of hemoglobin. Cells treated without a tested sample were used as no-lysis control. Cells treated with 10% Triton X-100 were used as complete lysis control. Data are presented as mean ± standard deviation ($n = 3$ biological replicates). The statistical significance of the data was assessed using one-way ANOVA followed by Tukey's multiple comparisons test. ****$p < 0.001$ vs. 10% Triton X-100-treated cells. **d** Counts of various blood cells 7 days after that of LUN@RBP$_{P545}$ and UPSN@CBD$_{SA97}$ administration. WBC, white blood cells; RBC, red blood cells; PLT, platelets. Data are presented as mean ± standard deviation ($n = 3$ biological replicates). The

statistical significance of the data was assessed using one-way ANOVA followed by Tukey's multiple comparisons test. ns, no significance. **e** Comprehensive blood chemistry panel taken 7 days after that of LUN@RBP$_{P545}$ and UPSN@CBD$_{SA97}$ administration. CRE, creatinine; BUN Blood urea nitrogen; TP Total protein, ALB albumin; ALT alanine transaminase, AST Aspartate transaminase, ALP Alkaline phosphatase. Data are presented as mean ± standard deviation ($n = 3$ biological replicates). The statistical significance of the data was assessed using one-way ANOVA followed by Tukey's multiple comparisons test. ns, no significance. **f** Body weight changes were recorded after treatment with LUN@RBP$_{P545}$, UPSN@CBD$_{SA97}$, or PBS over 7 d. Data are presented as mean ± standard deviation ($n = 10$ biological replicates). **g** Haematoxylin and eosin staining of histology sections from major organs 7 days after the intravenous administration of LUN@RBP$_{P545}$ and UPSN@CBD$_{SA97}$. Scale bars, 20 µm. Independent experiments ($n = 3$ biological replicates) were performed with similar results. Source data are provided as a Source Data file.

higher efficacy of Rif@LUN@RBP$_{P545}$ against CRKP in this model. A 100% recovery rate and long-term survival could be achieved by treating mice with 5 or 10 mg/kg of Rif@LUN@RBP$_{P545}$ (Fig. 7b). This result sharply contrasts the only 50% and 40% survival rate observed for treatment with 5 mg/kg of Rif@LUN and free rifampicin. The treated mice were further studied by bacterial load measurements in the lungs and other major organs, which confirmed the excellent therapeutic efficacy of Rif@LUN@RBP$_{P545}$ (Fig. 7c–g). The rifampicin loaded nanodelivery system significantly reduced the populations of CRKP in the lungs and other major organs of the infected mice.

The infection model for assessing the therapeutic efficacy of Rif@UPSN@CBD$_{SA97}$ was devised in a similar method, this time intratracheally introducing MRSA. The resulting pneumonia caused 80% mortality ($n = 10$ mice) 12–48 h post-infection (Fig. 7a, h). Treatment of the infected mice with intravenous injections of Rif@UPSN@CBD$_{SA97}$ at a dose range of 0.3–2.7 mg/kg 12 h post-infection resulted in 100% recovery and long-term survival (Fig. 7h). Treatment with similar doses of free rifampicin resulted in an only 40–70% survival rate (Supplementary Fig. 8). To achieve a 90% survival rate, 8.1 mg/kg of rifampicin had to be administered (Supplementary Fig. 8), compared to 0.3 mg/kg of Rif@UPSN@CBD$_{SA97}$, demonstrating a 27-fold higher efficacy of the nanodelivery system compared to the free antibiotic in this model. The excellent therapeutic efficacy of Rif@UPSN@CBD$_{SA97}$ was confirmed by bacterial load measurements in the lungs and other major organs of infected mice, which showed that Rif@UPSN@CBD$_{SA97}$ significantly reduced the populations of MRSA in the investigated organs (Fig. 7i–m). The results from both delivery systems tested here confirm that RBPs and CBDs have great potential as targeting modules in nanodelivery systems. In the form of antibiotic delivery systems, they have already proven to be highly efficient against difficult to treat infections with ESKAPE pathogens.

### Nanodelivery systems improve the therapeutic efficacy of antibiotics against resistant pathogens

Finally, LUN@RBP$_{P545}$ and UPSN@CBD$_{SA97}$ were used to improve the therapeutic efficacy of antibiotics to which the target bacteria are resistant. Imipenem-loaded LUN@RBP$_{P545}$ (Imi@LUN@RBP$_{P545}$, with an imipenem loading efficiency of 58% by mass) and ampicillin-loaded UPSN@CBD$_{SA97}$ (Amp@UPSN@CBD$_{SA97}$, with an ampicillin loading efficiency of 56% by mass) were engineered as described for Rif@LUN@RBP$_{P545}$ and Rif@UPSN@CBD$_{SA97}$, respectively. An infection model to test Imi@LUN@RBP$_{P545}$ efficacy was constructed by intratracheally introducing CRKP, resulting pneumonia caused 90% mortality ($n = 10$ mice) 12–48 h post-infection (Fig. 8a). Treatment of the infected mice with intravenous injections of Imi@LUN@RBP$_{P545}$ at a dose of 10 mg/kg 12 h post-infection resulted in 80% recovery and long-term survival (Fig. 8a), whereas an identical dose by weight of free imipenem showed a 10% survival rate like the untreated group

(Supplementary Fig. 9). Notably, administration of imipenem at a dose of 320 mg/kg was required to equal the 90% recovery achieved by 20 mg/kg Imi@LUN@RBP$_{P545}$ (Fig. 8a, Supplementary Fig. 9), demonstrating a 16-fold higher efficacy of Imi@LUN@RBP$_{P545}$ compared to the free antibiotic in this model. In addition, treatment of the infected mice with a dose of 20 mg/kg free imipenem was ineffective, and treatment with 20 mg/kg Imi@LUN led to 30% recovery (Fig. 8a). A 100% recovery rate and long-term survival could be achieved by treating mice with 40 mg/kg Imi@LUN@RBP$_{P545}$ (Fig. 8a). The ability of Imi@LUN@RBP$_{P545}$ to improve the therapeutic efficacy of imipenem in CRKP was further confirmed by bacterial load determinations in the lungs and other major organs of infected mice. This assay showed that Imi@LUN@RBP$_{P545}$ significantly reduced the populations of CRKP in the investigated organs (Fig. 8b–f).

In the meantime, an infection model was created by intratracheally introducing MRSA that leads to 90% mortality 12–48 h post infection. Treatment of the infected mice with intravenous injections of Amp@UPSN@CBD$_{SA97}$ at a dose of 10 mg/kg 12 h post-infection resulted in 90% recovery and long-term survival (Fig. 8g), whereas an identical dose by weight of free ampicillin was ineffective with only a 10% survival rate (Supplementary Fig. 10). Moreover, administration of ampicillin at a dose of 320 mg/kg was required to equal the 90% recovery achieved by 10 mg/kg Amp@UPSN@CBD$_{SA97}$ (Fig. 8g, Supplementary Fig. 10), demonstrating a 32-fold higher efficacy of Amp@UPSN@CBD$_{SA97}$ against MRSA in this model. Notably, treatment of the infected mice with doses of 20 mg/kg and 40 mg/kg Amp@UPSN@CBD$_{SA97}$ resulted in 100% recovery and long-term survival. This result sharply contrasts the only 20–30% survival rate observed for treatment with free ampicillin at doses of 20 mg/kg and 40 mg/kg (Fig. 8g, Supplementary Fig. 10). The treated mice were further studied by bacterial load determinations in the lungs and other major organs, which confirms the ability of Amp@UPSN@CBD$_{SA97}$ to improve the effectiveness of ampicillin in MRSA. The ampicillin-loaded UPSN@CBD$_{SA97}$ significantly reduced the populations of MRSA in the lungs and other major organs of the infected mice (Fig. 8h–l). Together, the results from both Imi@LUN@RBP$_{P545}$ and Amp@UPSN@CBD$_{SA97}$ demonstrate that RBPs and CBDs-guided nanodelivery systems have excellent potential for improving the therapeutic efficacy of antibiotics against resistant pathogens.

In summary, in this study, two antibiotic nanodelivery systems were developed that significantly increased the therapeutic efficacy of rifampicin, imipenem, and ampicillin against two ESKAPE pathogens. By employing bacteria-targeting proteins sourced from bacteriophages, silica nanoparticles loaded with antibiotics could be delivered to in vivo infection sites with high specificity. Using the respective targeting systems, a 16-fold increase in rifampicin and imipenem efficacy is demonstrated in vivo against the carbapenem-resistant Gram-negative *K. pneumoniae*, and a 27 to 32-fold increase in rifampicin and

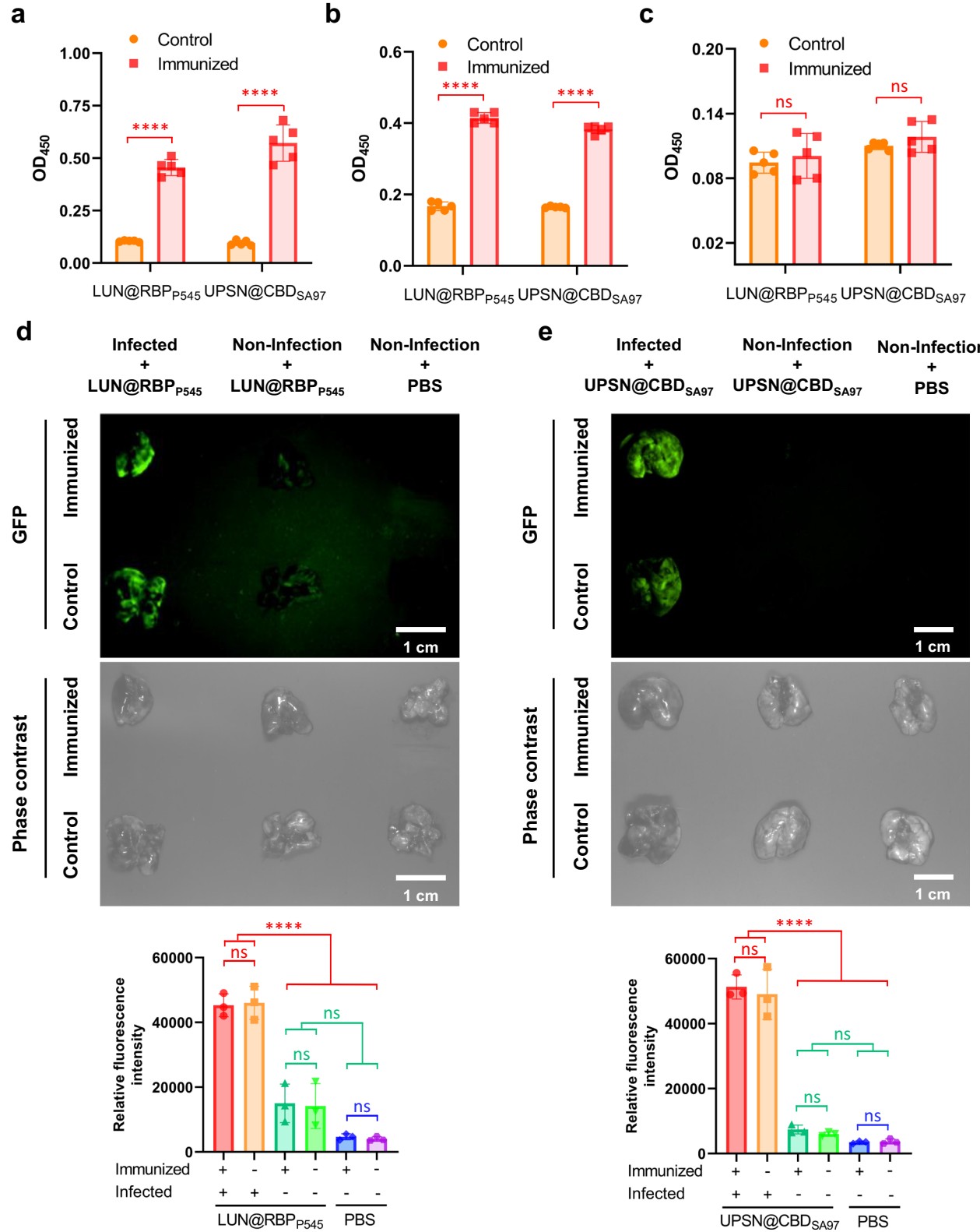

ampicillin efficacy is demonstrated in vivo against the methicillin-resistant Gram-positive *S. aureus*. Next to their increased activity, the phage-protein nanodelivery systems developed in this study have several advantages over free antibiotic and previously developed nanodelivery systems. Since these systems are guided by targeting-proteins from bacteriophages, the system can be modified to target any bacteria by replacing the targeting proteins with those from naturally occurring phages. Additionally, since the phage proteins are

target specific, they concentrate at the infection site. This means that less antibiotic is needed, and a lower concentration than administered will be present in the rest of the body, limiting side effects. All of the previous combined with the excellent biocompatibility demonstrated for LUN@RBP$_{P545}$ and UPSN@CBD$_{SA97}$, phage-protein nanodelivery systems are a promising strategy to increase antibiotic efficacy against both Gram-positive and Gram-negative bacteria. And, since this strategy is modular, keeping it up to date should be much cheaper than the

**Fig. 6 | LUN@RBP$_{P545}$ and UPSN@CBD$_{SA97}$ are re-applicable.** Levels of RBP$_{P545}$-specific and CBD$_{SA97}$-specific antibodies, IgG (**a**), IgM (**b**), and IgA (**c**), after immunization of LUN@RBP$_{P545}$ and UPSN@CBD$_{SA97}$, respectively, for twice. Antibody titers for RBP$_{P545}$ and CBD$_{SA97}$ in mouse serum were determined by ELISA using 1000-fold diluted samples (IgG) or 100-fold diluted samples (IgM and IgA). Data are presented as mean ± standard deviation ($n$ = 5 biological replicates). The statistical significance of the data was assessed using one-way ANOVA followed by Tukey's multiple comparisons test. ns, no significance; ****$p < 0.0001$. **d** Time-gated fluorescence image of LUN@RBP$_{P545}$, in which RBP$_{P545}$ was fused with GFP (green), in lungs harvested from mice after 30 min of circulation. LUN@RBP$_{P545}$ immunized and PBS-treated mice were intravenously inoculated with *K. pneumoniae* to generate the CRKP-induced lung infection mouse model. At 24 h post-infection, LUN@RBP$_{P545}$ was intravenously injected and allowed to circulate for 30 min. After that, lungs were harvested for time-gated fluorescence imaging using a FUSION FX7 EDGE Imaging System. Mice without CRKP infection were treated with the same dose of LUN@RBP$_{P545}$ or the same volume of PBS as controls. Data are presented as mean ± standard deviation ($n$ = 3 biological replicates). The statistical significance of the data was assessed using one-way ANOVA followed by Tukey's multiple comparisons test. ns, no significance; ****$p < 0.0001$. **e** Time-gated fluorescence image of UPSN@CBD$_{SA97}$, in which CBD$_{SA97}$ was fused with GFP (green), in lungs harvested from mice after 30 min of circulation. UPSN@CBD$_{SA97}$ immunized and PBS-treated mice were intravenously inoculated with *S. aureus* to generate the MRSA-induced lung infection mouse model. At 24 h post-infection, UPSN@CBD$_{SA97}$ was intravenously injected and allowed to circulate for 30 min. After that, lungs were harvested for time-gated fluorescence imaging using a FUSION FX7 EDGE Imaging System. Mice without MRSA infection were treated with the same dose of UPSN@CBD$_{SA97}$ or the same volume of PBS as controls. Data are presented as mean ± standard deviation ($n$ = 3 biological replicates). The statistical significance of the data was assessed using one-way ANOVA followed by Tukey's multiple comparisons test. ns, no significance; ****$p < 0.0001$. Source data are provided as a Source Data file.

continuous discovery and development of novel antibiotics against difficult-to-treat infections.

## Methods

### Ethics statement

All animal experiments conformed to the Guide for the Care and Use of Laboratory Animals from the National Institutes of Health, and all procedures were approved by the Animal Research Committee of Sichuan Agricultural University [permission number 20230074]. The use of New Zealand white rabbit biological materials (erythrocytes isolated from the blood of healthy rabbits) for research was approved by Sichuan Agricultural University Institution Review Board.

### Bacterial strains used and growth conditions

Strains and plasmids used in this study are listed in Supplementary Tables 3, 7. *E. coli* TOP10 chemical competent cells were used as hosts in the construction of all plasmids. *E. coli* BL21(DE3) chemical competent cells were transformed with the verified plasmids and used for subsequent expression of the plasmids encoding proteins. For plasmid selection, *E. coli* strains were grown in Luria–Bertani (LB) medium or on LB medium solidified with 1% (wt/vol) agar at 37 °C, supplemented with 20 μg/mL kanamycin for selection purposes. For protein expression, stationary-phase cultures, which were grown in LB, were inoculated (50-fold diluted) on Tryptic Soy Broth (TSB) and induced with Isopropyl β-d-1-thiogalactopyranoside (IPTG) (0.5 mM) at OD$_{600}$ = 0.6. All indicator strains were inoculated on LB and incubated at 37 °C with aeration at 220 rpm for preparing the overnight cultures.

### Animals

Six-week old mice (SPF-grade ICR, female) were purchased from Chengdu Dossy Experimental Animals Co., Ltd. All mice were housed in the Animal Center of the College of Veterinary Medicine, Sichuan Agricultural University under standard conditions with free access to food and water. The light was from 8:00 am to 8:00 pm, with the temperature kept at 22 ± 1 °C and humidity at 40–70%. All experimental procedures involving animals were in accordance with the guidelines of the Animal Care and Use Committee of Sichuan Agricultural University. All animal experiments were performed independently of each other with different cohorts of mice.

### Molecular biology techniques

Oligonucleotide primers used for cloning and sequencing in this study are listed in Supplementary Table 8, and all the oligonucleotide primers and oligonucleotide inserts were purchased from Chengdu Youkangjianxing Biotechnology Co., Ltd. (Chengdu, China). *RBP$_{P545}$* and *CBD$_{SA97}$* genes were inserted into *pRSFDuet-1* that contains a *His6-tagged GFP* gene, to generate *pRSF-His6-GFP-RBP$_{P545}$* and *pRSF-His6-GFP-CBD$_{SA97}$*, respectively, using ClonExpress Ultra One Step Cloning

Kit (Cat No. C115-02, Vazyme Biotech Co., Ltd., Nanjing, China). Subsequently, a cysteine residue gene was cloned into *pRSF-His6-GFP-RBP$_{P545}$* and *pRSF-His6-GFP-CBD$_{SA97}$*, at the N-terminus of these recombinant proteins, by amplifying template plasmid using downstream sense- and upstream antisense primers with a Cys-encoding tail and ca. 15 bp overlap on the 5′, to generate *pRSF-Cys-His6-GFP-RBP$_{P545}$* and *pRSF-Cys-His6-GFP-CBD$_{SA97}$*, respectively. DNA amplification was carried out using 2 × Phanta Max Master Mix (Dye Plus) (Cat No. P525-02, Vazyme Biotech Co., Ltd., Nanjing, China). The designed plasmids were verified by sequencing at Chengdu Youkangjianxing Biotechnology Co., Ltd. (Chengdu, China).

### Expression, purification, and characterization of His6-tagged proteins

*E. coli* BL21(DE3) cells were transformed with the *pRSF-Cys-His6-GFP-RBP$_{P545}$* and *pRSF-Cys-His6-GFP-CBD$_{SA97}$* plasmids (30 ng), respectively, plated on LB agar plates containing 20 μg/mL kanamycin, and grown at 37 °C for 18 h. A single colony of each of these plates was used to inoculate 20 mL of LB supplemented with 20 μg/mL kanamycin, and grown for 16 h at 37 °C with aeration at 220 rpm. After that, the culture was used to inoculate 1 L (50-fold dilution) of TSB supplemented with 20 μg/mL kanamycin. Cultures were grown at 37 °C to an OD$_{600}$ of 0.6. The cultures were chilled in ice water for 10 min, after which protein expression was induced by the addition of IPTG to a final concentration of 0.5 mM, and the cultures were grown at 18 °C for 24 h with aeration at 220 rpm.

After that, the cultures were centrifuged at 6000 g for 15 min, and the cell pellets were collected. The pellets were resuspended in lysis buffer (50 mM Tris-HCl, 2 mM EDTA, 100 mM NaCl, 0.5% Triton X-100, pH 8.5), and the suspension was sonicated for 20 min in total. The insoluble material was subsequently removed by centrifugation at 12,000 g for 20 min, and supernatants were filtered through a 0.45 μm membrane. The supernatants were applied to Ni-NTA agarose columns (Beijing Solarbio & Technology Co., ltd., Beijing, China) equilibrated with 50 mM NaH$_2$PO$_4$, 500 mM NaCl, 10 mM imidazole, pH 8.0. The flow-through was discarded, and the column was subsequently washed with 15 CV of wash buffer (50 mM NaH$_2$PO$_4$, 500 mM NaCl, 20 mM imidazole, pH 8.0). The proteins were eluted with 6 CV elution buffer (50 mM NaH$_2$PO$_4$, 500 mM NaCl, 500 mM imidazole, pH 8.0). After that, the proteins were further purified by a GE prepacked gel filtration column (HiPrep™ 16/60 Sephacryl® S-200 HR, GE17-1166-01).

Last, the purified gRBP$_{P545}$ and gCBD$_{SA97}$ were separated by 8% and 12% SDS-PAGE, respectively, and visualized by Coomassie Blue staining. In addition, western blotting assays were performed to specifically identify the purified products by employing 1:20000 diluted His-Tag (6*His) Monoclonal antibody and 1:5000 diluted HRP-conjugated Affinipure Goat Anti-Mouse IgG(H + L).

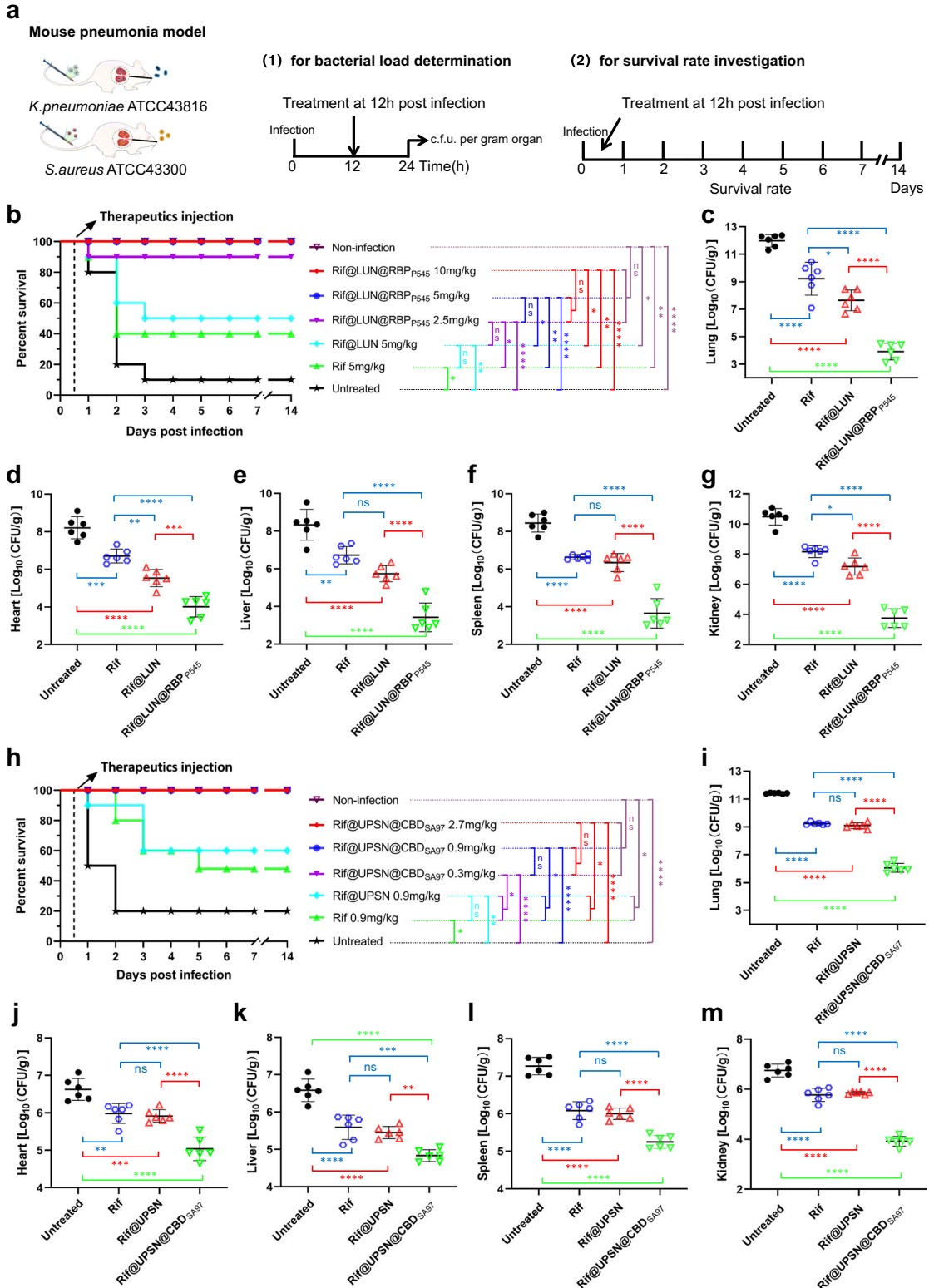

## Characterization of antibiotic-loaded nanodelivery systems

**Dynamic light scattering and zeta potential measurements.** The hydrodynamic diameters and zeta potentials of UPSN, UPSN-NH$_2$, Rif@UPSN, Rif@UPSN@CBD$_{SA97}$, Rif@LUN, and Rif@LUN@RBP$_{P545}$ (in ultrapure water) were measured using a Malvern Zetasizer Nano ZSE (Malvern Instruments).

**Transmission electron microscopy (TEM).** A TEM instrument (JEM-2100plus, Japan) operating at 120 kV accelerating voltage was used to

record TEM images. The samples were prepared by drop-casting 5 μL of sample onto a carbon-coated copper grid (200 Mesh, Beijing XXBR Technology CO., Ltd.). The sample was dried for at least 1 h before TEM imaging.

**Confocal laser scanning microscopy.** To identify whether the engineered gCBD$_{SA97}$ and gRBP$_{P545}$ were correctly conjugated on Rif@UPSN and Rif@LUN, respectively, in a visualization way, confocal laser scanning microscopy assays were performed. In

**Fig. 7 | In vivo therapeutic efficacies of the two developed distinct bacterial targeted precise antibiotic nanodelivery systems, Rif@LUN@RBP$_{P545}$ and Rif@UPSN@CBD$_{SA97}$. a** Scheme of the experimental protocol for the mouse pneumonia models. **b** Survival rates of mice in the CRKP-induced mouse pneumonia model ($n = 10$ biological replicates). Survival was analyzed by the Log-rank (Mantel-Cox) test. ns, no significance; *$p < 0.05$; **$p < 0.01$; ****$p < 0.0001$. **c**−**g** Treated with Rif@LUN@RBP$_{P545}$ (5 mg/kg) significantly reduced the bacterial load of organs of the CRKP-induced pneumonia mouse relative to equivalent doses of untargeted rifampicin nanoparticles or of free rifampicin. At 24 h post-infection, the mice ($n = 6$) were euthanized by cervical dislocation. Bacterial loads (Log10 c.f.u. per gram of *K. pneumoniae*) of the lung (**c**), heart (**d**), liver (**e**), spleen (**f**), and kidney (**g**) were counted. Data are presented as mean ± standard deviation ($n = 6$ biological replicates). The statistical significance of the data was assessed using one-way ANOVA followed by Tukey's multiple comparisons test. ns, no significance;

*$p < 0.05$; **$p < 0.01$; ***$p < 0.001$; ****$p < 0.0001$. **h** Survival rates of mice in the MRSA-induced mouse pneumonia model ($n = 10$ biological replicates). Survival was analyzed by the Log-rank (Mantel-Cox) test. ns, no significance; *$p < 0.05$; **$p < 0.01$; ****$p < 0.0001$. **i**−**m** Treated with Rif@UPSN@CBD$_{SA97}$ (0.9 mg/kg) significantly reduced the bacterial load of organs of the MRSA-induced pneumonia mouse relative to equivalent doses of untargeted rifampicin nanoparticles or of free rifampicin. At 24 h post-infection, the mice ($n = 6$) were euthanized by cervical dislocation. Bacterial loads (Log10 c.f.u. per gram of *S. aureus*) of the lung (**i**), heart (**j**), liver (**k**), spleen (**l**), and kidney (**m**) were counted. Data are presented as mean ± standard deviation ($n = 6$ biological replicates). The statistical significance of the data was assessed using one-way ANOVA followed by Tukey's multiple comparisons test. ns, no significance; **$p < 0.01$; ***$p < 0.001$; ****$p < 0.0001$. Source data are provided as a Source Data file.

---

brief, UPSN-NH$_2$ was labeled with the red fluorescent dye DyLight 633 via an amine-NHS reaction. UPSN-NH$_2$ (10 mg) was suspended in 1 mL of ultrapure water containing 5 µg/mL of DyLight™ 633 NHS Easter, and the reaction was performed at room temperature in the dark for 1 h with stirring. Subsequently, the DyLight 633-labeled UPSN-NH$_2$ were collected and washed with ultrapure water three times before being applied for preparation of DyLight 633-labeled Rif@UPSN@CBD$_{SA97}$ and Rif@LUN@RBP$_{P545}$. Finally, the DyLight 633-labeled Rif@UPSN@CBD$_{SA97}$ and Rif@LUN@RBP$_{P545}$ were visualized by a STELLARIS STED/EM CPD300 confocal microscope (Leica, Germany).

### Bacterial pathogen binding capacity investigation
**Fluorescence microscopy assays.** The bacterial pathogen binding capacities of gRBP$_{P545}$ against *K. pneumoniae* strains and gCBD$_{SA97}$ against *S. aureus* strains were investigated by using fluorescence microscopy. In addition, to identify the binding specificity, the binding capacities of gRBP$_{P545}$ and gCBD$_{SA97}$ against other members of ESKAPE pathogens were investigated too. Briefly, the cell density of pathogens was normalized to an OD$_{600}$ of 0.2 in PBS, and RBP$_{P545}$ or CBD$_{SA97}$ was added to a final concentration of 20 µg/mL. The mixtures were incubated at 37 °C for 30 min. After washing the cell suspensions with PBS three times, the samples were loaded on 1.5% agarose pads and analyzed with a Nikon 80i (Japan) microscope.

**Confocal laser scanning microscopy assays.** The bacterial pathogen binding capacities of Rif@LUN@RBP$_{P545}$ and RBP$_{P545}$ against CRKP (ATCC 43816) and Rif@UPSN@CBD$_{SA97}$ and CBD$_{SA97}$ against MRSA (ATCC 43300) were investigated by using confocal laser scanning microscopy. After normalization of the cell density to an OD$_{600}$ of 0.2 in Mueller Hinton broth (MHB), pathogens cells were treated with DAPI at a final concentration of 1 mg/mL at 37 °C for 2 h. Subsequently, pathogens cells were treated with RBP$_{P545}$ (20 µg/mL), DyLight 633-labeled Rif@LUN@RBP$_{P545}$ (200 µg/mL), CBD$_{SA97}$ (20 µg/mL), or DyLight 633-labeled Rif@UPSN@CBD$_{SA97}$ (200 µg/mL) at 37 °C for 30 min. After washing the cell suspensions with PBS three times, the samples were loaded on 1.5% agarose pads and analyzed by STELLARIS STED/EM CPD300 confocal microscope (Leica, Germany).

### gRBP$_{P545}$ and gCBD$_{SA97}$-mediated targeting to infection sites
**K. pneumoniae-induced mouse pneumonia model.** SPF-grade ICR mice (female, 6 weeks, 20 ± 2 g, $n = 3$ per group) were infected intratracheally with CRKP (ATCC 43816) at a dose of $6 \times 10^9$ c.f.u. per mouse. At 24 h post-infection, mice were treated with gRBP$_{P545}$ (0.5 mg per mouse) or Rif@LUN@RBP$_{P545}$ (4 mg per mouse) via intravenous injection. Mice without CRKP infection were treated with gRBP$_{P545}$ (0.5 mg per mouse), Rif@LUN@RBP$_{P545}$ (4 mg per mouse), or 0.1 M PBS as controls. After circulation for 30 min, the mice were sacrificed, and the lungs were collected. Finally, the fluorescence images of the lungs were captured by using a FUSION FX7 EDGE Imaging System.

**S. aureus-induced mouse pneumonia model.** SPF-grade ICR mice (female, 6 weeks, 20 ± 2 g, $n = 3$ per group) were infected intratracheally with MRSA (ATCC 43300) at a dose of $6 \times 10^9$ c.f.u. per mouse. At 24 h post-infection, mice were treated with gCBD$_{SA97}$ (0.5 mg per mouse) or Rif@UPSN@CBD$_{SA97}$ (4 mg per mouse) via intravenous injection. Mice without MRSA infection were treated with gCBD$_{SA97}$ (0.5 mg per mouse), Rif@UPSN@CBD$_{SA97}$ (4 mg per mouse), or 0.1 M PBS as controls. After circulation for 30 min, the mice were sacrificed, and the lungs were collected. Last, the fluorescence images of the lungs were captured by using a FUSION FX7 EDGE Imaging System.

### Mouse pneumonia infections and treatments
**Therapeutic efficacy of Rif@LUN@RBPP545 in a CRKP-induced mouse pneumonia model.** To assess the in vivo antibacterial efficacy of the engineered Rif@LUN@RBP$_{P545}$, SPF-grade ICR mice (female, 6 weeks, 20 ± 2 g, $n = 10$ per group) were infected intratracheally with CRKP (ATCC 43816) at a dose ($8 \times 10^9$ c.f.u. per mouse) that leads to 80% mortality 12–48 h post infection. At 12 h post-infection, mice were treated with Rif@LUN@RBP$_{P545}$ (10 mg/kg), Rif@LUN@RBP$_{P545}$ (5 mg/kg), Rif@LUN@RBP$_{P545}$ (2.5 mg/kg), Rif@LUN (5 mg/kg), free rifampicin (5 mg/kg), or 0.1 M PBS via intravenous injection. Mice without CRKP infection were used as the non-infection control. The survival rates of different groups were monitored for 14 days.

To get a deeper insight into the antibacterial efficacy of Rif@LUN@RBP$_{P545}$ in vivo, SPF-grade ICR mice (female, 6 weeks, 20 ± 2 g, $n = 6$ per group) were infected intratracheally with CRKP (ATCC 43816) at a dose ($8 \times 10^9$ c.f.u. per mouse). At 12 h post-infection, mice were treated with Rif@LUN@RBP$_{P545}$ (5 mg/kg), Rif@LUN (5 mg/kg), free rifampicin (5 mg/kg), or 0.1 M PBS via intravenous injection. Mice without CRKP infection were used as the non-infection control. At 24 h post-infection, organs, including heart, liver, spleen, lung, and kidney, were collected to measure the bacterial load.

**Therapeutic efficacy of Rif@UPSN@CBDSA97 in a MRSA-induced mouse pneumonia model.** To assess the in vivo antibacterial efficacy of the engineered Rif@UPSN@CBD$_{SA97}$, SPF-grade ICR mice (female, 6 weeks, 20 ± 2 g, $n = 10$ per group) were infected intratracheally with MRSA (ATCC 43300) at a dose ($1 \times 10^{10}$ c.f.u. per mouse) that leads to 80% mortality 12–48 h post infection. At 12 h post-infection, mice were treated with Rif@UPSN@CBD$_{SA97}$ (2.7 mg/kg), Rif@UPSN@CBD$_{SA97}$ (0.9 mg/kg), Rif@UPSN@CBD$_{SA97}$ (0.3 mg/kg), Rif@UPSN (0.9 mg/kg), free rifampicin (0.9 mg/kg), or 0.1 M PBS via intravenous injection. Mice without MRSA infection were used as the non-infection control. The survival rates of different groups were monitored for 14 days.

To get a deeper insight into the antibacterial efficacy of Rif@UPSN@CBD$_{SA97}$ in vivo, SPF-grade ICR mice (female, 6 weeks, 20 ± 2 g, $n = 6$ per group) were infected intratracheally with MRSA (ATCC 43300) at a dose ($1 \times 10^{10}$ c.f.u. per mouse) that leads to 80% mortality 12–48 h post infection. At 12 h post-infection, mice were treated with Rif@UPSN@CBD$_{SA97}$ (0.9 mg/kg), Rif@UPSN (0.9 mg/kg),

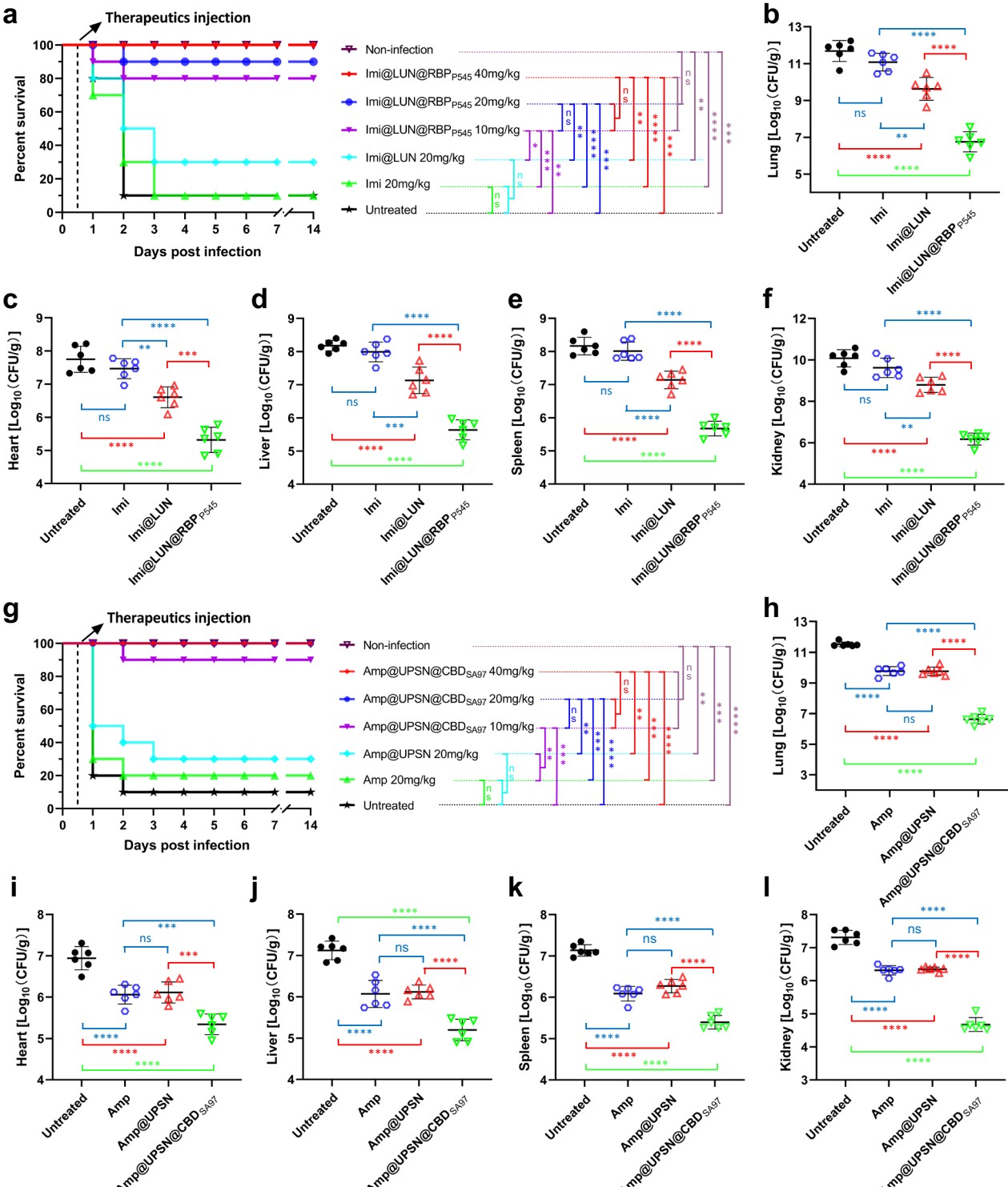

**Fig. 8 | In vivo therapeutic efficacies of Imi@LUN@RBP$_{P545}$ and Amp@UPSN@CBD$_{SA97}$ against resistant pathogens. a** Survival rates of mice in the CRKP-induced mouse pneumonia model ($n = 10$ biological replicates). Survival was analyzed by the Log-rank (Mantel-Cox) test. ns, no significance; *$p < 0.05$; **$p < 0.01$; ***$p < 0.001$; ****$p < 0.0001$. **b–f** Treated with Imi@LUN@RBP$_{P545}$ (20 mg/kg) significantly reduced the bacterial load of organs of the CRKP-induced pneumonia mouse relative to equivalent doses of untargeted imipenem nanoparticles or of free imipenem. At 24 h post-infection, the mice ($n = 6$) were euthanized by cervical dislocation. Bacterial loads (Log10 c.f.u. per gram of *K. pneumoniae*) of the lung (**b**), heart (**c**), liver (**d**), spleen (**e**), and kidney (**f**) were counted. Data are presented as mean ± standard deviation ($n = 6$ biological replicates). The statistical significance of the data was assessed using one-way ANOVA followed by Tukey's multiple comparisons test. ns, no

significance; **$p < 0.01$; ***$p < 0.001$; ****$p < 0.0001$. **g** Survival rates of mice in the MRSA-induced mouse pneumonia model ($n = 10$ biological replicates). Survival was analyzed by the Log-rank (Mantel-Cox) test. ns, no significance; **$p < 0.01$; ***$p < 0.001$; ****$p < 0.0001$. **h–l** Treated with Amp@UPSN@CBD$_{SA97}$ (20 mg/kg) significantly reduced the bacterial load of organs of the MRSA-induced pneumonia mouse relative to equivalent doses of untargeted ampicillin nanoparticles or of free ampicillin. At 24 h post-infection, the mice ($n = 6$) were euthanized by cervical dislocation. Bacterial loads (Log10 c.f.u. per gram of *S. aureus*) of the lung (**h**), heart (**i**), liver (**j**), spleen (**k**), and kidney (**l**) were counted. Data are presented as mean ± standard deviation ($n = 6$ biological replicates). The statistical significance of the data was assessed using one-way ANOVA followed by Tukey's multiple comparisons test. ns, no significance; ***$p < 0.001$; ****$p < 0.0001$. Source data are provided as a Source Data file.

free rifampicin (0.9 mg/kg), or 0.1 M PBS via intravenous injection. Mice without MRSA infection were used as the non-infection control. At 24 h post-infection, organs, including heart, liver, spleen, lung, and kidney, were collected to measure the bacterial load.

## Statistical analysis

All the statistical analyses were performed using GraphPad Prism 8 software (GraphPad Software). All data represent mean value ± standard deviation. The statistical significance of the data was assessed using one-way ANOVA followed by Tukey's multiple comparisons test with GraphPad Prism 8.0. Survival was analyzed by the Log-rank (Mantel-Cox) test with GraphPad Prism 8.0. ns, no significance; *$p < 0.05$; **$p < 0.01$; ***$p < 0.001$; ****$p < 0.0001$.

## Reporting summary

Further information on research design is available in the Nature Portfolio Reporting Summary linked to this article.

## Data availability

The authors declare that all data supporting the findings of this study are available within the paper and its Supplementary Information/ Source Data file. Source data are provided with this paper.

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

## Acknowledgements

We thank Prof. Qihui Zhou (School of Rehabilitation Sciences and Engineering, University of Health and Rehabilitation Sciences, Qingdao, China.) for providing the clinically isolated *K. pneumoniae* strain. We thank Prof. Yanglei Yi (College of Food Science and Engineering, Northwest A&F University, Yangling, Shaanxi, China.) for providing the clinically isolated *S. aureus* strains. This work was supported by the National Key Research and Development Program (2023YFB3809902, Y.Liu), the "1000-Talent Program" in Sichuan Province (2287, X.Zhao; 1923, H.W.), the Science and Technology Project of Sichuan Province (2022YFH0057, X.Zhao; 2022YFH0062, H.W.; 2022ZYD0068, H.W.), and the National Natural Science Foundation of China (32102689, H.W.).

## Author contributions

X.Zhao and H.W. conceived the project and strategies. H.W., Z.Y., and Y. Liu supervised the work and corrected the manuscript. X.Zhao, X.Zhong, and S.Y. designed and carried out the experiments, analyzed data, and wrote the manuscript. K.D. did the confocal laser scanning microscopy assays. J.D., Z.H., and Y.Li did experimental work on therapeutic efficacy studies. J.H.V. and H.W. wrote the manuscript and analyzed data. All authors contributed to and commented on the manuscript text and approved its final version.

## Competing interests

The authors declare no competing interests.
