## [Peer Review file · Nature Communications]

REVIEWER COMMENTS

Reviewer #1 (Remarks to the Author):

Review of Guiding antibiotics towards their target using bacteriophage proteins.

Summary.

Using bacteriophage proteins that are involved in attachment (RBP) and lysis (CBD) specific for two different bacteria, in the paper of *K. pneumoniae* and *S. aureus* the author proposed a nanodelivery system to combat the rise of antimicrobial resistance.

The authors explore the use of these nanodrugs to specifically target the bacterium of choice. Clearly showing that the system designed for the Gram Negative bacteria is not effective in targeting the Gram positive and vis a versa. They do this in vitro, with and without the antibiotic delivery mechanism and in vivo again with and without the delivery mech. Biosafety evaluation are carried out showing minimal cytotoxic effects. The paper is summarized with work showing greater survivability from using the guided antibiotics compared to an equal concentration of free antibiotic.

The paper is well written however suffers several flaws which reduces my enthusiasm. Without addressing those flaws I will not be recommending it for publication in Nature Communications.

First, there is use of an obvious experiment to see if these drugs work on antibiotic resistant bacteria. The authors introduction makes it clear they are aware of the issue, but I cannot think why they would not have tried this treatment on the actual problem. They should test these methods on antibiotic resistant isolates. None of the isolates chosen 5 KP or 8 SA are clearly resistant to Rif.

Second, there is a lack of immune response investigation. Phage therapy has been shown to generate an immune response. I would assume these molecules likewise will generate immune responses. The authors should investigate these. Are antibodies produced and amplified on reapplication? What cytokines etc. are produced?

Third, there is a lack of statistical analysis of the survivability assays. There are several easy to apply stats tests which should be applied to the survival data.

Some minor comments:

The introduction mentions covid, you could replace this with any co infection.

Explaining how the RBP and CBDs function in a normal viral lifecycle will help guide readers as to why they make good targets for this type of research.

On page 7 it is mentioned that 5 strains of *K. pneumoniae* and eight *S. aureus* strains were chosen. Why these 5 and 8? I could not find any justification for them.

Bottom of page 13 “RBP-mediated lipid bilayer-coated nanodelivery system demonstrated here, is theoretically less toxic and therefore has better potential for clinical applications”

Prove this, cite it, or remove it.

Reviewer #2 (Remarks to the Author):

The article entitled “Guiding antibiotics towards their target using bacteriophage proteins”, authored by Xinghong Zhao, Xinyi Zhong, Shinong Yang, Jiarong Deng, Kai Deng, Zhengqun Huang, Yuanfeng Li, Zhongqiong Yin, Yong Liu, Jakob H. Viel, and Hongping Wan describes novel drug delivery system, which utilizes phage-derived proteins as target-recognition elements. The authors showed an impressive increase in the efficacy of rifampicin distributed using nanocarriers, and a free drug to treat acute lung infections in vivo was examined. Nanocarriers themselves were proven to be biocompatible. The article is coherent, describe an interesting story and important results. I believe it might be published in Nature Communications after major revision.

The list of the issues to be reconsidered and improved is as follows (without any particular order):

1. The Authors stated that “the microscopy images show efficient binding and cellular uptake”. I don’t find compelling evidence for the uptake. It is believed that bacteria (both G+ and G-) are unlikely to uptake such large particles. A great body of literature, both newer (e.g., <https://doi.org/10.1002/nano.202200049>) or a bit older (e.g., <https://doi.org/10.1128/AEM.71.5.2548-2557.2005>), claims that particles of up to few nanometers can be somehow easily internalized, with probability of such events decreasing with increasing size. In the described case, the particles are >150 nm, and even a proper lipid envelope might not be enough to facilitate the process. In Figure 1C, the authors drew a nanocarrier being internalized by fusing it with the bacterium's membrane. But the cell envelope in G- consists of a peptidoglycan layer and a second internal membrane. How do the carriers pass through them? There is not enough data to support this claim. The only possible piece of evidence is the microscopy picture in Figure 4a (right column, first (G-) vs third (G+) row). Such an image might be a consequence of particles being on top or beneath the cell. More experiments are needed to support the claim. Stating that it is similar to the case reported in ref. 51 is, in my opinion, not enough.

2. Some parts of the manuscript are very “soft” and lack hard data. For example, abstract should be more specific. Please provide exact numbers to support sentences like “the nanodelivery systems suppressed pathogen infections more effectively than higher doses of free antibiotic”. In another example, the Authors stated that “nanodelivery systems have comparable or even better bacterial targeting ability than the already impressive CARG peptide-guided nanodelivery system” and it is not clear how this was assessed.

3. Materials and methods are somehow described in a way that is hard to reproduce. For example, pH is crucial in the coupling reaction of gCBDSA97 to Rif@UPSN. If pH is higher than 8, the reaction can occur on both ends of the polymer uncontrolled. The other example is DyLight staining – it is unclear if the stained material was used for in vivo studies. It is also unclear whether labeling affected the loading capacity of the antibiotic (by, for instance, occupying some space in the UPSN particles). Yet another example of this problem is that there is no information given on how to store these agents to prevent antibiotics from leaching before administration and, at the same time, not to affect the targeting biomolecules. The release protocol in SI suggests storage as powder, but please elaborate, especially in the context of protein stability. Such small details are crucial to ensure the reproducibility of the results.

4. DLS results in Fig S6 do not correspond to Table S3. The errors are larger than those given in the table. The Authors should describe how they collected and analyzed DLS data.

5. The 50 nm shown in the TEM picture is too much to correspond to the lipid bilayer. The structure of the lipid envelope seems different.

6. There are still some typos and unfortunate statements, e.g., “Scale bars are donated to the images”, “of 1 mg/mL-1.” (SI), “at 12,000× g for 20 min”, “ 10×10^9 c.f.u. per mouse” should be written as 10^{10} . Some abbreviations, e.g., DSPE, PEG, BCA assay, TEA, are not introduced.

7. The protocol for BCA assay is not described.

8. Ethical statements are provided in the middle of the methods section, which seems odd.

9. In the Materials and Methods section “Mouse pneumonia infections and treatments,” only a single concentration is mentioned, but in the text (and Figures S7 and S8), the concentrations of the given agent varied. Please comment or correct.

Reviewer #3 (Remarks to the Author):

Manuscript "Guiding antibiotics towards their target using bacteriophage proteins" by Zhao et al is a research article describing the novel strategy of targeted antibiotic delivery with high therapeutic potential aimed to increase concentration of applied antibiotic especially at infection site in case of the treatment difficult to treat infections caused by ESKAPE pathogens (*Klebsiella pneumoniae* and *Staphylococcus aureus*). The work presented for evaluation has a significant contribution to the development of the contemporary medicine; targeted treatment of bacterial infections, especially in the era of ineffectiveness of antibiotics due to the spread of antibiotic resistance mechanisms among bacteria and difficulties in achieving therapeutic concentration or penetration of antibiotics into the site of infection may constitute a breakthrough. The manuscript contain well organized and comprehensively described study. In my opinion the experiments conducted in the manuscript were designed appropriately, data analysis, interpretation of results was carried out correctly, conclusions are also appropriate to the obtained results. Overall the article is well-organized with up to dated reference.

Detailed comments:

1. I suggest arranging the keywords in alphabetical order.
2. Introduction: line 54 please complete data on recently discovered antibiotics.
3. Results and Discussion, line 136, SDS-PAGE- please expand this abbreviation.

Reviewer #1

Review of Guiding antibiotics towards their target using bacteriophage proteins.

Summary.

Using bacteriophage proteins that are involved in attachment (RBP) and lysis (CBD) specific for two different bacteria, in the paper of *K. pneumoniae* and *S. aureus* the author proposed a nanodelivery system to combat the rise of antimicrobial resistance.

The authors explore the use of these nanodrugs to specifically target the bacterium of choice. Clearly showing that the system designed for the Gram Negative bacteria is not

effective in targeting the Gram positive and vis a versa. They do this in vitro, with and without the antibiotic delivery mechanism and in vivo again with and without the delivery mech. Biosafety evaluation are carried out showing minimal cytotoxic effects. The paper is summarized with work showing greater survivability from using the guided antibiotics compared to an equal concentration of free antibiotic.

The paper is well written however suffers several flaws which reduces my enthusiasm. Without addressing those flaws I will not be recommending it for publication in Nature Communications.

Response:

We thank the reviewer for these supportive comments and constructive criticism. The additional experiments you suggested made a lot of sense, and we were happy to perform them to substantiate our claims. We did additional infection studies which showed that the engineered nanodelivery systems enhanced the therapeutic efficacies of imipenem and ampicillin against carbapenem-resistant *K. pneumoniae* and methicillin-resistant *S. aureus* infections, respectively. In addition, extra immunization and time-gated fluorescence imaging assays showed that the nanodelivery systems could be applied at least thrice without losing effectiveness (although an immune response is present). We have analyzed the survival data by the Log-rank (Mantel-Cox) test with GraphPad Prism 8.0, and all issues the reviewer raised have been responded to and revised accordingly.

Q1. First, there is use of an obvious experiment to see if these drugs work on antibiotic resistant bacteria. The authors introduction makes it clear they are aware of the issue, but I cannot think why they would not have tried this treatment on the actual problem. They should test these methods on antibiotic resistant isolates. None of the isolates chosen 5 KP or 8 SA are clearly resistant to Rif.

Response:

You make a fair point here!

Antibiotic resistance usually comes in the form of (drastically) decreased sensitivity to a specific antibiotic. Therefore, a more than tenfold increase in activity of any antibiotic can also be expected to increase the efficacy of antibiotics to which the strain is resistant. However, we completely agree that performing the actual experiments significantly substantiates this hypothesis.

We have done these experiments, and the results showed that both imipenem-loaded LUN@RBP_{P545} and ampicillin-loaded UPSN@CBD_{SA97} showed good therapeutic efficacy in our carbapenem-resistant *K. pneumoniae*-induced mouse pneumonia model and methicillin-resistant *S. aureus*-induced mouse pneumonia model, respectively. In contrast, the free imipenem and ampicillin were ineffective. These results demonstrate that RBPs and CBDs-guided nanodelivery systems have the potential to improve the therapeutic efficacy of antibiotics against resistant pathogens. Please see the revised manuscript for more details.

Q2. Second, there is a lack of immune response investigation. Phage therapy has been shown to generate an immune response. I would assume these molecules likewise will generate immune responses. The authors should investigate these. Are antibodies produced and amplified on reapplication? What cytokines etc. are produced?

Response:

We thank the reviewer for this constructive suggestion, which greatly improved the quality of our manuscript. We have done extra immunization assays and time-gated fluorescence imaging assays to assess the reapplication potential of the engineered nanodelivery systems.

The results obtained from the immunization assays demonstrate that LUN@RBP_{P545} and UPSN@CBD_{SA97} have shown immunogenicity in mice, RBP_{P545}-specific and CBD_{SA97}-specific antibodies, IgG and IgM, were produced. Interestingly, further time-gated fluorescence imaging analysis demonstrates that the infection site targeting capabilities of LUN@RBP_{P545} and UPSN@CBD_{SA97} do not decrease in repeat administration. Both LUN@RBP_{P545} and UPSN@CBD_{SA97} caused an immune response *in vivo*, but the produced antibodies did not decrease the targeting abilities of nanodelivery systems. While the effect of more than three applications has to be further researched, the initial results look promising.

Q3. Third, there is a lack of statistical analysis of the survivability assays. There are several easy to apply stats tests which should be applied to the survival data.

Response:

We thank the reviewer for this valuable suggestion! We have analyzed the survival data by the Log-rank (Mantel-Cox) test with GraphPad Prism 8.0, and correlation analyses were evaluated by Pearson r^2 , ns: $p > 0.05$, * $p < 0.05$, ** $p < 0.01$, *** $p < 0.001$, and **** $p < 0.0001$. Please see the revised manuscript and its SI for details.

Some minor comments:

Q4. The introduction mentions covid, you could replace this with any co infection.

Response:

We have revised the sentence to “As a result, important medical treatments that rely on antibiotics, like organ transplants, chemotherapy, or prevention of co-infection, are predicted to become riskier and less successful in the future”.

Q5. Explaining how the RBP and CBDs function in a normal viral lifecycle will help guide readers as to why they make good targets for this type of research.

Response:

The functions of RBPs and CBDs have been introduced to some extent in the introduction section. However, we agree that a broader description of the bacteriophage lifecycle would put the use of RBP and CBD into a broader perspective. Because this paper is already of quite a technical nature, we felt that a more in-depth description of the bacteriophage life cycle would risk distracting the reader from the main story. However, for the interested reader, we have added citations of good reviews discussing the role of RBP and CBD in the phage life cycle. Please see References 24, 27-36, especially Ref. 24, for the role of RBP and CBD in the phage life cycle.

Q6. On page 7 it is mentioned that 5 strains of *K. pneumoniae* and eight *S. aureus* strains were chosen. Why these 5 and 8? I could not find any justification for them.

Response:

We agree that the choice of strains is not well described in the main text, and adding their sources would overcomplicate the text. Because of this, we have changed the description to: “After purification, the binding capacity of gRBP_{P545} and gCBD_{SA97} was verified by respectively incubating them with different *K. pneumoniae* and *S. aureus* strains (Supplementary Table 3).” We obtained one carbapenem-resistant *K. pneumoniae* standard strain, two carbapenem-resistant *K. pneumoniae* clinical isolates, and two carbapenem-sensitive standard strains, in total five strains, to investigate the binding capacity of gRBP_{P545}. In addition, two methicillin-resistant *S. aureus* standard strains, three methicillin-sensitive *S. aureus* standard strains, and one methicillin-resistant and two methicillin-sensitive *S. aureus* clinical isolates (in total eight strains) were obtained for assessing the binding capacity of gRBP_{P545}. Although the specific number of strains is not that crucial, we felt like testing a broader panel of different strains would substantiate our claims and would possibly bring to light aberrations in the results for specific target strains. This being said, two of the most

important antibiotic-resistant pathogen classes, carbapenem-resistant *K. pneumoniae* and methicillin-resistant *S. aureus* were involved. These showed that, especially with the new experimental data, our nanomedicines also target antibiotic-resistant pathogens. A detailed list of used strains can be found in Supplementary Table 3.

Q7. Bottom of page 13 “RBP-mediated lipid bilayer-coated nanodelivery system demonstrated here, is theoretically less toxic and therefore has better potential for clinical applications”

Prove this, cite it, or remove it.

Response:

We have removed this statement from the revised manuscript, as you and the reviewer #2 suggested. Thanks!

Reviewer #2

The article entitled “Guiding antibiotics towards their target using bacteriophage proteins”, authored by Xinghong Zhao, Xinyi Zhong, Shinong Yang, Jiarong Deng, Kai Deng, Zhengqun Huang, Yuanfeng Li, Zhongqiong Yin, Yong Liu, Jakob H. Viel, and Hongping Wan describes novel drug delivery system, which utilizes phage-derived proteins as target-recognition elements. The authors showed an impressive increase in the efficacy of rifampicin distributed using nanocarriers, and a free drug to treat acute lung infections in vivo was examined. Nanocarriers themselves were proven to be biocompatible. The article is coherent, describe an interesting story and important results. I believe it might be published in Nature Communications after major revision.

Response:

Thanks a lot for your supportive comments and good suggestions. We have thoroughly revised the manuscript according to your suggestions. In addition, we have done additional experiments, as the reviewer #1 suggested.

The list of the issues to be reconsidered and improved is as follows (without any particular order):

Q1. The Authors stated that “the microscopy images show efficient binding and cellular uptake”. I don’t find compelling evidence for the uptake. It is believed that bacteria (both G+ and G-) are unlikely to uptake such large particles. A great body of literature, both newer (e.g., <https://doi.org/10.1002/nano.202200049>) or a bit older (e.g., <https://doi.org/10.1128/AEM.71.5.2548-2557.2005>), claims that particles of up

to few nanometers can be somehow easily internalized, with probability of such events decreasing with increasing size. In the described case, the particles are >150 nm, and even a proper lipid envelope might not be enough to facilitate the process. In Figure 1C, the authors drew a nanocarrier being internalized by fusing it with the bacterium's membrane. But the cell envelope in G- consists of a peptidoglycan layer and a second internal membrane. How do the carriers pass through them? There is not enough data to support this claim. The only possible piece of evidence is the microscopy picture in Figure 4a (right column, first (G-) vs third (G+)row). Such an image might be a consequence of particles being on top or beneath the cell. More experiments are needed to support the claim. Stating that it is similar to the case reported in ref. 51 is, in my opinion, not enough.

Response:

Thank you for pointing out this important issue! After carefully reading the relevant articles, we completely agree that the particles described in the manuscript are unlikely to be taken up by the bacteria. While our initial claim was inspired by other literature (Wu, S. et al. Bacterial outer membrane-coated mesoporous silica nanoparticles for targeted delivery of antibiotic rifampicin against Gram-negative bacterial infection in vivo. Adv. Funct. Mater. 31, 2103442 (2021)), we should have done our due diligence here. Consequently, we have removed this statement from the revised manuscript.

Q2. Some parts of the manuscript are very “soft” and lack hard data. For example, abstract should be more specific. Please provide exact numbers to support sentences like “the nanodelivery systems suppressed pathogen infections more effectively than higher doses of free antibiotic”. In another example, the Authors stated that “nanodelivery systems have comparable or even better bacterial targeting ability than the already impressive CARG peptide-guided nanodelivery system” and it is not clear how this was assessed.

Response:

We have changed “the nanodelivery systems suppressed pathogen infections more effectively than higher doses of free antibiotic” to “the nanodelivery systems suppressed pathogen infections more effectively than 16 to 32-fold higher doses of free antibiotics”.

In addition, we agree with the reviewer that there is no hard data that can support the statement “nanodelivery systems have comparable or even better bacterial targeting ability than the already impressive CARG peptide-guided nanodelivery system”. The idea behind the statement is based on the large natural reservoir of phages and their

targets, which does not exist for cyclic peptides. However, we agree with you that we cannot substantiate this claim, so this statement has been removed from the revised manuscript.

Q3. Materials and methods are somehow described in a way that is hard to reproduce. For example, pH is crucial in the coupling reaction of gCBDSA97 to Rif@UPSN. If pH is higher than 8, the reaction can occur on both ends of the polymer uncontrolled. The other example is DyLight staining – it is unclear if the stained material was used for *in vivo* studies. It is also unclear whether labeling affected the loading capacity of the antibiotic (by, for instance, occupying some space in the UPSN particles). Yet another example of this problem is that there is no information given on how to store these agents to prevent antibiotics from leaching before administration and, at the same time, not to affect the targeting biomolecules. The release protocol in SI suggests storage as powder, but please elaborate, especially in the context of protein stability. Such small details are crucial to ensure the reproducibility of the results.

Response:

We thank the reviewer for pointing this out! We have added the reaction buffer information to the materials and methods section (in SI). The reaction buffer used for both amine-NHS reaction and thiol-maleimide reaction was 0.1 M phosphate buffer at pH 7.4, which was conducted under the guidelines of Thermo Scientific for crosslink reactions [Scientific, Thermo. "Thermo scientific crosslinking technical handbook." Waltham (USA): Thermo Scientific (2012).].

The DyLight (An expensive chemical) stained nanoparticles were only used in the confocal laser scanning microscopy assays. We have mentioned DyLight staining only in confocal laser scanning microscopy studies; the UPSN was not stained for the *in vivo* studies. This is clear to the readers.

Yes, the nanomedicines generated in this study should be stored as lyophilized powder at -20 °C, and sterilized sodium chloride solution or phosphate-buffered saline solution is recommended to prepare the nanomedicine resuspensions for administration. We have added this information to the materials and methods section (in SI).

Q4. DLS results in Fig S6 do not correspond to Table S3. The errors are larger than those given in the table. The Authors should describe how they collected and analyzed DLS data.

Response:

We thank the reviewer for pointing out this! The Fig. S6 is showing the hydrodynamic **size distribution** of the nanoparticles measured by DLS. However, the data in Table S4

(Table S3 in the previous version) is showing the **Z-average size** (average hydrodynamic diameter) of the nanoparticles from three experimental replications. We have changed the “hydrodynamic diameter” to “Z-average size” to make it more clear. (Please see Table S3).

Q5. The 50 nm shown in the TEM picture is too much to correspond to the lipid bilayer. The structure of the lipid envelope seems different.

Response:

We have added scale bars for the zoomed photos (upper left corner) in Fig. 3b. The observed thickness (about 20nm) of the lipid envelope is consistent with previous studies (please see references below), which showed a lipid envelope thickness ranging from 10 to 50nm.

Ref.1: Wu, Shuang, Yi Huang, Jiachang Yan, Yuzhen Li, Jinfeng Wang, Yi Yan Yang, Peiyan Yuan, and Xin Ding. "Bacterial outer membrane-coated mesoporous silica nanoparticles for targeted delivery of antibiotic rifampicin against Gram-negative bacterial infection in vivo." *Advanced Functional Materials* 31, no. 35 (2021): 2103442.

Ref.2: Kim, Byungji, Hong-Bo Pang, Jinyoung Kang, Ji-Ho Park, Erkki Ruoslahti, and Michael J. Sailor. "Immunogene therapy with fusogenic nanoparticles modulates macrophage response to *Staphylococcus aureus*." *Nature communications* 9, no. 1 (2018): 1969.

Q6. There are still some typos and unfortunate statements, e.g., “Scale bars are donated to the images”, “of 1 mg/mL-1.” (SI), “at 12,000× g for 20 min”, “10×10⁹ c.f.u. per mouse” should be written as 10¹⁰. Some abbreviations, e.g., DSPE, PEG, BCA assay, TEA, are not introduced.

Response:

Thank you for pointing out these (admittedly sometimes embarrassing) mistakes. We have made the necessary corrections, which are listed below.

We have removed “Scale bars are donated to the images” from the Fig. 4 legend since the length of scale bars is shown in the pictures.

We have corrected “of 1 mg/mL-1.” (“of 1 mg/mL.”) accordingly in the SI.

We have replaced “at 12,000× g for 20 min” with “at 12,000 g for 20 min” in the revised manuscript.

We have changed “10×10⁹ c.f.u. per mouse” to “1×10¹⁰ c.f.u. per mouse” accordingly.

The full names of DSPE-PEG₂₀₀₀ (1,2-distearoyl-sn-glycero-3-phosphoethanolamine-N-[amino(polyethylene glycol)-2000]), DSPE-PEG₂₀₀₀-MAL (1,2-distearoyl-sn-glycero-3-phosphoethanolamine-N-[maleimide(polyethylene glycol)-2000]), PEG (polyethylene glycol), BCA (Bicinchoninic acid assay), and TEA (Triethylamine) have been added to the revised manuscript and its SI, where the first time they present in the manuscript or SI. In addition, we have checked and revised other similar issues, such as SDS-PAGE (sodium dodecyl sulfate-polyacrylamide gel electrophoresis).

Q7. The protocol for BCA assay is not described.

Response:

We have added the protocol for BCA assay to the materials and methods section (SI). Please see below the detailed protocol:

The quantity of gCBD_{SA97} and gRBP_{P545} on the nano vehicles was verified by a BCA assay. The bicinchoninic acid assay was performed according to the manufacturer's guidelines (Cat No. PC0020, Beijing Solarbio & Technology Co., Ltd., Beijing, China). Briefly, the standard protein bovine serum albumin (BSA) was diluted with PBS at a series of concentrations of 2000, 1500, 1000, 750, 500, 250, 125, and 25 µg/mL. After treating the samples (20 µL) with BCA working solution (200 µL) in a 96-well plate for 30 min at 37 °C, the absorbance values were measured using a Thermo Scientific Varioskan Flash multimode microplate reader at a wavelength of 562 nm. The amount of gCBD_{SA97} and gRBP_{P545} on the nano vehicles was calculated using the BSA standard protein as a reference.

Q8. Ethical statements are provided in the middle of the methods section, which seems odd.

Response:

We have changed "Ethical statement" to "Animals", in which the ethical statement is involved.

Q9. In the Materials and Methods section "Mouse pneumonia infections and treatments," only a single concentration is mentioned, but in the text (and Figures S7 and S8), the concentrations of the given agent varied. Please comment or correct.

Response:

To assess the therapeutic effects of the nanoparticles, pre-experiments were performed with free antibiotic treatments to find suitable doses that show about a 50% survival rate. This is the reason that the data in Fig. S7 and S8 includes six free antibiotic doses. After that, the therapeutic effects of the nanoparticles (three different doses)

were assessed by survival assays. Finally, to get deeper insight into the therapeutic effects of the nanoparticles, a single dose of the nanoparticles was used for the bacterial load assays.

Reviewer #3

Manuscript "Guiding antibiotics towards their target using bacteriophage proteins" by Zhao et al is a research article describing the novel strategy of targeted antibiotic delivery with high therapeutic potential aimed to increase concentration of applied antibiotic especially at infection site in case of the treatment difficult to treat infections caused by ESKAPE pathogens (*Klebsiella pneumoniae* and *Staphylococcus aureus*). The work presented for evaluation has a significant contribution to the development of the contemporary medicine; targeted treatment of bacterial infections, especially in the era of ineffectiveness of antibiotics due to the spread of antibiotic resistance mechanisms among bacteria and difficulties in achieving therapeutic concentration or penetration of antibiotics into the site of infection may constitute a breakthrough. The manuscript contain well organized and comprehensively described study. In my opinion the experiments conducted in the manuscript were designed appropriately, data analysis, interpretation of results was carried out correctly, conclusions are also appropriate to the obtained results. Overall the article is well-organized with up to dated reference.

Response:

Thank you for your kind words and constructive comments! We have implemented all your suggestions into our manuscript.

Detailed comments:

Q1. I suggest arranging the keywords in alphabetical order.

Response:

Thanks! We have rearranged the keywords in alphabetical order.

Q2. Introduction: line 54 please complete data on recently discovered antibiotics.

Response:

We have updated the data of newly approved antibiotics to 2023 (García-Castro, M., Sarabia, F., Díaz-Morilla, A. & López-Romero, J. M. Approved antibacterial drugs in the last 10 years: From the bench to the clinic. *Explor. Drug Sci.* 1, 180–209 (2023).), including novel antibiotics in clinical trials (Walesch, S. et al. Fighting antibiotic resistance-strategies and (pre) clinical developments to find new antibacterials. *EMBO*

Rep. 24, e56033 (2023).). There were no new antibiotics approved by the FDA in 2023 (<https://www.fda.gov/drugs/new-drugs-fda-cders-new-molecular-entities-and-new-therapeutic-biological-products/novel-drug-approvals-2023>).

Q3. Results and Discussion, line 136, SDS-PAGE- please expand this abbreviation.

Response:

Thank you for pointing this out! We have added the full name of SDS-PAGE (sodium dodecyl sulfate-polyacrylamide gel electrophoresis) to the revised manuscript. In addition, we have checked and revised other similar issues.

REVIEWERS' COMMENTS

Reviewer #1 (Remarks to the Author):

The authors have addressed my comments adequately. I hope they feel as I do that the addition of the immune work up and testing on resistant isolates strengthens their work.